# Riemannian Diffusion Models

**Chin-Wei Huang**\*, **Milad Aghajohari**\*, **Avishek Joey Bose**
**Prakash Panangaden, Aaron Courville**
University of Montreal & McGill University & Mila
{chin-wei.huang, milad.aghajohari, aaron.courville}@umontreal.ca
joey.bose@mail.mcgill.ca, prakash@cs.mcgill.ca

## Abstract

Diffusion models are recent state-of-the-art methods for image generation and likelihood estimation. In this work, we generalize continuous-time diffusion models to arbitrary Riemannian manifolds and derive a variational framework for likelihood estimation. Computationally, we propose new methods for computing the Riemannian divergence which is needed for likelihood estimation. Moreover, in generalizing the Euclidean case, we prove that maximizing this variational lower-bound is equivalent to Riemannian score matching. Empirically, we demonstrate the expressive power of Riemannian diffusion models on a wide spectrum of smooth manifolds, such as spheres, tori, hyperboloids, and orthogonal groups. Our proposed method achieves new state-of-the-art likelihoods on all benchmarks.

## 1 Introduction

By learning to transmute noise, generative models seek to uncover the underlying generative factors that give rise to observed data. These factors can often be cast as inherently geometric quantities as the data itself need not lie on a flat Euclidean space. Indeed, in many scientific domains such as high-energy physics (Brehmer & Cranmer, 2020), directional statistics (Mardia & Jupp, 2009), geoscience (Mathieu & Nickel, 2020), computer graphics (Kazhdan et al., 2006), and linear biopolymer modeling such as protein and RNA (Mardia et al., 2008; Boomsma et al., 2008; Frellsen et al., 2009), data is best represented on a Riemannian manifold with a *non-zero curvature*. Naturally, to effectively capture the generative factors of these data, we must take into account the geometry of the space when designing a learning framework.

Recently, diffusion based generative models have emerged as an attractive model class that not only achieve likelihoods comparable to state-of-the-art autogressive models (Kingma et al., 2021) but match the sample quality of GANs without the pains of adversarial optimization (Dhariwal & Nichol, 2021). Succinctly, a diffusion model consists of a fixed Markov chain that progressively transforms data to a prior defined by the inference path, and a generative model which is another Markov chain that is learned to invert the inference process (Ho et al., 2020; Song et al., 2021b).

While conceptually simple, the learning framework can have a variety of perspectives and goals. For example, Huang et al. (2021) provide a variational framework for general continuous-time diffusion processes on Euclidean manifolds as well as a functional Evidence Lower Bound (ELBO) that can be equivalently shown to be minimizing an implicit score matching objective. At present, however, much of the success of diffusion based generative models and its accompanying variational framework is purpose built for Euclidean spaces, and more specifically, image data. It does not easily translate to general Riemannian manifolds.

In this paper, we introduce Riemannian Diffusion Models (RDM)—generalizing conventional diffusion models on Euclidean spaces to arbitrary Riemannian manifolds. Departing from diffusion models on Euclidean spaces, our approach uses the Stratonovich SDE formulation for which the

36th Conference on Neural Information Processing Systems (NeurIPS 2022).

conventional chain rule of calculus holds, which, as we demonstrate in section §3, can be exploited to define diffusion on a Riemannian manifold. Furthermore, we take an extrinsic view of geometry by defining the Riemannian manifold of interest as an embedded sub-manifold within a higher dimensional (Euclidean) ambient space. Such a choice enables us to define both our inference and generative SDEs using the coordinate system of the ambient space, greatly simplifying the implementation of the theory developed using the intrinsic view.

**Main Contributions**. We summarize our main contributions below:

- We introduce a variational framework built on the Riemannian Feynman-Kac representation and Giransov's theorem. In Theorem 2 we derive a Riemannian continuous-time ELBO, strictly generalizing the CT-ELBO in Huang et al. (2021) and prove in Theorem 4 that its maximization is equivalent to Riemannian score matching for marginally equivalent SDEs (Theorem 3).
- To compute the Riemannian CT-ELBO it is necessary to compute the Riemannian divergence of our parametrized vector field, for which we introduce a QR-decomposition-based method that is computationally efficient for low dimensional manifolds as well a projected Hutchinson method for scalable unbiased estimation. Notably, our approach does not depend on the closest point projection which may not be freely available for many Riemannian manifolds of interest.
- We also provide a variance reduction technique to estimate the Riemannian CT-ELBO objective that leverages importance sampling with respect to the time integral, which crucially avoids carefully designing the noise schedule of the inference process.
- Empirically, we validate our proposed models on spherical manifolds towards modelling natural disasters as found in earth science datasets, products of spherical manifolds (tori) for protein and RNA, synthetic densities on hyperbolic spaces and orthogonal groups. Our empirical results demonstrate that RDM leads to new state-of-art likelihoods over prior manifold generative models.

## 2 Background

In this section, we provide the necessary background on diffusion models and key concepts from Riemannian geometry that we utilize to build RDMs. For a short review of the latter, see Appendix A or Ratcliffe (1994) for a more comprehensive treatment of the subject matter.

### 2.1 Euclidean diffusion models

A diffusion model can be defined as the solution to the (Itô) SDE (Øksendal, 2003),

$$\mathrm{d}X = \mu \, \mathrm{d}t + \sigma \, \mathrm{d}B_t, \tag{1}$$

with the initial condition $X_0$ following some unstructured prior $p_0$ such as the standard normal distribution, where $B_t$ is a standard Brownian motion, and $\mu$ and $\sigma$ are the drift and diffusion coefficients of the diffusion process, which control the deterministic forces driving the evolution and the amount of noise injected at each time step. This provides us a way to sample from the model, via numerically solving the dynamics from $t = 0$ to $t = T$ for some fixed termination time $T$. To train the model via maximum likelihood, we require an expression for the log marginal density of $X_T$, denoted by $\log p(x, T)$, which is generally intractable.

The marginal likelihood can be represented using a stochastic instantaneous change-of-variable formula, by applying the Feynman-Kac theorem to the Fokker-Planck PDE of the density. An application of Girsanov's theorem followed by an application of Jensen's inequality leads to the following variational lower bound (Huang et al., 2021; Song et al., 2021a):

$$\log p(x, T) \geq \mathbb{E}\left[\log p_0(Y_T) - \int_0^T \left(\frac{1}{2} \|a(Y_s, s)\|_2^2 + \nabla \cdot \mu(Y_s, T - s)\right) \mathrm{d}s \,\middle|\, Y_0 = x\right] \tag{2}$$

where $a$ is the variational degree of freedom, $\nabla \cdot$ denotes the (Euclidean) divergence operator, and $Y_s$ follows the inference SDE (the generative coefficients are evaluated in reversed time, *i.e.* $T - s$)

$$\mathrm{d}Y = (-\mu + \sigma a) \, \mathrm{d}s + \sigma \, \mathrm{d}\hat{B}_s \tag{3}$$

with $\hat{B}_s$ being another Brownian motion. This is known as the continuous-time evidence lower bound, or the CT-ELBO for short.

## 2.2 Riemannian manifolds

We work with a $d$-dimensional Riemannian manifold $(\mathcal{M}, g)$ embedded in a higher dimensional ambient space $\mathbb{R}^m$, for $m > d$. This assumption does not come with a loss of generality, since any Riemannian manifold can be isometrically embedded into a Euclidean space by the *Nash embedding theorem* (Gunther, 1991). In this case, the metric $g$ coincides with the pullback of the Euclidean metric by the inclusion map. Now, given a coordinate chart $\varphi : \mathcal{M} \to \mathbb{R}^d$ and its inverse $\psi = \varphi^{-1}$, we can define $\tilde{E}_j$ for $j = 1, \cdots, d$ to be the basis vectors of the tangent space $\mathcal{T}_x\mathcal{M}$ at point $x \in \mathcal{M}$. The tangent space can be understood as the pushforward of the Euclidean derivation of the patch space along $\psi$; *i.e.*, for any smooth function $f \in C^\infty(\mathcal{M})$, $\tilde{E}_j(f) = \frac{\partial}{\partial \tilde{x}_j} f \circ \psi$.

We denote by $P_x$ the orthogonal projection onto the linear subspace spanned by the column vectors of the Jacobian $J_x = \mathrm{d}\psi/\mathrm{d}\tilde{x}$. Specifically, $P_x$ can be constructed via $P_x = J_x(J_x^T J_x)^{-1} J_x^T$. Note that this subspace is isomorphic to the tangent space $\mathcal{T}_x\mathcal{M}$, which itself is a subspace of $\mathcal{T}_x\mathbb{R}^m$. As a result, we identify this subspace with $\mathcal{T}_x\mathcal{M}$. Lastly, we refer to the action of $P_x$ as the projection onto the tangential subspace, and $P_x$ itself as the tangential projection.

## 2.3 SDE on manifolds

Unlike Euclidean spaces, Riemannian manifolds generally do not possess a vector space structure. This prevents the direct application of the usual (stochastic) calculus. We can resolve this by defining the process via test functions. Specifically, let $V_k$ be a family of smooth vector fields on $\mathcal{M}$, and let $Z^k$ be a family of semimartingales (Protter, 2005). Symbolically, we write

$$\mathrm{d}X_t = \sum_k V_k(X_t) \circ \mathrm{d}Z_t^k \quad \text{if} \quad \mathrm{d}f(X_t) = \sum_k V_k(f)(X_t) \circ \mathrm{d}Z_t^k \tag{4}$$

for any $f \in C^\infty(\mathcal{M})$ (Hsu, 2002). The $\circ$ in the second differential equation is to be interpreted in the Stratonovich sense (Protter, 2005). The use of the Stratonovich integral is the first step deviating from the Euclidean diffusion model (1), as the Itô integral does not follow the usual chain rule.

Working with this abstract definition is not always convenient, so instead we work with specific coordinates of $\mathcal{M}$. Let $\varphi$ be a chart, and let $\tilde{v} = (\tilde{v}_{jk})$ be a matrix representing the coefficients of $V_k$ in the coordinate basis—i.e. $V_k(f) = \sum_{j=1}^d \tilde{v}_{jk} \frac{\partial}{\partial \tilde{x}_j} f \circ \varphi^{-1}\big|_{\tilde{x}=\varphi(x)}$. This allows us to write $\mathrm{d}\varphi(X_t) = \tilde{v} \circ \mathrm{d}Z$. Similarly, suppose $\mathcal{M}$ is a submanifold embedded in $\mathbb{R}^m$, and denote by $v = (v_{ik})$ the coefficients wrt the Euclidean basis. $v$ and $\tilde{v}$ are related by $v = \frac{\mathrm{d}\varphi^{-1}}{\mathrm{d}\tilde{x}} \tilde{v}$. Then we can express the dynamics of $X$ as a regular SDE using the Euclidean space's coefficients $\mathrm{d}X = v \circ \mathrm{d}Z$. Notably, by the relation between $v$ and $\tilde{v}$, the column vectors of $v$ are required to lie in the span of the column vectors of the Jacobian $\frac{\mathrm{d}\varphi^{-1}}{\mathrm{d}\tilde{x}}$ which restricts the dynamics to move tangentially on $\mathcal{M}$.

# 3 Riemannian diffusion models

We now develop a variational framework to estimate the likelihood of a diffusion model defined on a Riemannian manifold $(\mathcal{M}, g)$. Let $X_t \in \mathcal{M}$ be a process solving the following SDE:

$$\text{Generative SDE:} \qquad \mathrm{d}X = V_0 \, \mathrm{d}t + V \circ \mathrm{d}B_t, \qquad X_0 \sim p_0 \tag{5}$$

where $V_0$ and the columns of the diffusion matrix[1] $V := [V_1, \cdots, V_w]$ are smooth vector fields on $\mathcal{M}$, and $B_t$ is a $w$-dimensional Brownian motion. The law of the random variable $X_t$ can be written as $p(x, t) \, \mu(\mathrm{d}x)$, where $p(x, t)$ is the probability density function and $\mu$ is the $d$-dimensional Hausdorff measure on the manifold associated with the Riemannian volume density. Let $V \cdot \nabla$ be a differential operator defined by $(V \cdot \nabla_g)U := \sum_{k=1}^w (\nabla_g \cdot U_k)V_k$, where $\nabla_g \cdot U_k$ denotes the *Riemannian divergence* of the vector field $U_k$:

$$\nabla_g \cdot U_k = |G|^{-\frac{1}{2}} \sum_{j=1}^d \frac{\partial}{\partial \tilde{x}_j}(|G|^{\frac{1}{2}} \tilde{u}_{jk}). \tag{6}$$

Our first result is a stochastic instantaneous change-of-variable formula for the Riemannian SDE by applying the Feynman-Kac theorem to the Fokker Planck PDE of the density $p(x, t)$.

---

[1]The multiplication is interpreted similarly to matrix-vector multiplication, *i.e.* $V \circ \mathrm{d}B_t = \sum_{k=1}^w V_k \circ \mathrm{d}B_t^k$.

**Theorem 1** (**Marginal Density**). *The density $p(x, t)$ of the SDE (5) can be written as*

$$p(x,t) = \mathbb{E}\left[p_0\left(Y_t\right)\exp\left(-\int_0^t \nabla_g \cdot \left(V_0 - \frac{1}{2}(V \cdot \nabla_g)V\right)ds\right)\bigg| Y_0 = x\right] \qquad (7)$$

*where the expectation is taken wrt the following process induced by a Brownian motion $B'_s$*

$$dY = (-V_0 + (V \cdot \nabla_g)V)\,ds + V \circ dB'_s. \qquad (8)$$

For effective likelihood maximization, we require access to $\log p$ and its gradient. Towards this goal, we prove the following Riemannian CT-ELBO which serves as our training objective and follows from an application of change of measure (Girsanov's theorem) and Jensen's inequality.

**Theorem 2** (**Riemannian CT-ELBO**). *Let $\hat{B}_s$ be a $w$-dimensional Brownian motion, and let $Y_s$ be a process solving the following*

$$\text{Inference SDE:} \qquad dY = (-V_0 + (V \cdot \nabla_g)V + Va)\,ds + V \circ d\hat{B}_s, \qquad (9)$$

*where $a : \mathbb{R}^m \times [0, T] \to \mathbb{R}^m$ is the variational degree of freedom. Then we have*

$$\log p(x,T) \geq \mathbb{E}\left[\log p_0(Y_T) - \int_0^T \frac{1}{2}\|a(Y_s,s)\|_2^2 + \nabla_g \cdot \left(V_0 - \frac{1}{2}(V \cdot \nabla_g)V\right)ds \bigg| Y_0 = x\right],$$
$$(10)$$

*where all the generative degree of freedoms $V_k$ are evaluated in the reversed time direction.*

## 3.1 Computing Riemannian divergence

Similar to the Euclidean case, computing the Riemannian CT-ELBO requires computing the divergence "$\nabla_g \cdot$" of a vector field, which can be achieved by applying the following identity.

**Proposition 1** (**Riemannian divergence identity**). *Let $(M, g)$ be a $d$-dimensional Riemannian manifold. For any smooth vector field $V_k \in \mathfrak{X}(\mathcal{M})$, the following identity holds:*

$$\nabla_g \cdot V_k = \sum_{j=1}^d \left\langle \nabla_{\tilde{E}_j} V_k, \tilde{E}^j \right\rangle_g. \qquad (11)$$

*Furthermore, if the manifold is a submanifold embedded in the ambient space $\mathbb{R}^m$ equipped with the induced metric $g = \iota^* \bar{g}$, then*

$$(\nabla_g \cdot V_k)(x) = \text{tr}\left(P_x \frac{dv_k}{dx} P_x\right), \qquad (12)$$

*where $v_k = (v_{1k}, \cdots, v_{mk})$ are the ambient space coefficients $V_k = \sum_{i=1}^m v_{ik}\frac{\partial}{\partial x_i}$ and $P_x$ is the orthogonal projection onto the tangent space.*

**Intrinsic coordinates**. The patch-space formula (6) can be used to compute the Riemannian divergence. This view was adopted by Mathieu & Nickel (2020), where they combined the Hutchinson trace identity and the internal coordinate formula to estimate the divergence. The drawbacks of this framework include: (1) obtaining local coordinates may be difficult for some manifolds, hindering generality in practice; (2) we might need to change patches, which complicates implementations; and (3) the inverse scaling of $\sqrt{|G|}$ might result in numerical instability and high variance.

**Closest-point projection**. The coordinate-free expression (11) leads to the closest-point projection method proposed by Rozen et al. (2021). Concretely, define the closest-point projection by $\pi(x) := \arg\min_{y \in \mathcal{M}} \|x - y\|$, where $\|\cdot\|$ is the Euclidean norm. Let $V_k(x)$ be the derivation corresponding to the ambient space vector $v_k(x) = P_{\pi(x)}u(\pi(x))$ for some unconstrained $u : \mathbb{R}^m \to \mathbb{R}^m$. Rozen et al. (2021) showed that $\nabla_g \cdot V_k(x) = \nabla \cdot v_k(x)$, since $v_k$ is infinitesimally constant in the normal

direction to $\mathcal{T}_x\mathcal{M}$. This allows us to compute the divergence directly in the ambient space. However, the closest-point projection map $\pi$ may not always be easily obtained.

**QR decomposition**. An alternative to the closest-point projection is to instead search for an orthogonal basis for $\mathcal{T}_x\mathcal{M}$. Let $Q = [e_1, \cdots, e_d, n_1, \cdots, n_{m-d}]$ be an orthogonal matrix whose first $d$ columns span the $\mathcal{T}_x\mathcal{M}$, and the remaining $m - d$ vectors span its orthogonal complement $\mathcal{T}_x\mathcal{M}^\perp$. To construct $Q$ we can simply sample $d$ vectors—*e.g.* from $\mathcal{N}(0,1)$–in the ambient space and orthogonally project them to $\mathcal{T}_x\mathcal{M}$ using $P_x$. These vectors, although not orthogonal yet, form a basis for $\mathcal{T}_x\mathcal{M}$. Next we concatenate them with $m - d$ random vectors and apply a simple $QR$ decomposition to retrieve an orthogonal basis. Using $Q$ we may rewrite equation (12) as follows:

$$(\nabla_g \cdot V_k)(x) = \mathrm{tr}\left( QQ^\top P_x \frac{\mathrm{d}v_k}{\mathrm{d}x} P_x \right) = \mathrm{tr}\left( (P_x Q)^\top \frac{\mathrm{d}v_k}{\mathrm{d}x} P_x Q \right) = \sum_{j=1}^d e_j^\top \frac{\mathrm{d}v_k}{\mathrm{d}x} e_j \qquad (13)$$

where we used (1) the orthogonality of $Q$, (2) the cyclic property of trace, (3) and the fact that $P_x e_j = e_j$ and $P_x n_j = 0$. In practice, concatenation with the remaining $m - d$ vectors is not needed as they are effectively not used in computing the divergence, speeding up computation when $m \gg d$. Moreover, the vector-Jacobian product can be computed in $\mathcal{O}(m)$ time using reverse-mode autograd and importantly does not require the closest-point projection $\pi$.

**Projected Hutchinson**. When QR is too expensive for higher dimensional problems, the Hutchinson trace estimator (Hutchinson, 1989) can be employed within the extrinsic view representation (12). For example, let $z$ be a standard normal vector (or a Rademacher vector), we have $(\nabla_g \cdot V_k)(x) = \mathbb{E}_{z \sim \mathcal{N}, z' = P_x z}[z'^\top \frac{\mathrm{d}v_k}{\mathrm{d}x} z']$. Different from a direct application of the trace estimator to the closest-point method, we directly project the random vector to the tangent subspace. Therefore, the closest-point projection is again not needed.

### 3.2 Fixed-inference parameterization

Following prior work (Sohl-Dickstein et al., 2015; Ho et al., 2020; Huang et al., 2021), we let the inference SDE (9) be defined as a simple noise process taking observed data to unstructured noise:

$$\mathrm{d}Y = U_0 \,\mathrm{d}t + V \circ \mathrm{d}\hat{B}_s, \qquad (14)$$

where $U_0 = \frac{1}{2}\nabla_g \log p_0$ and $V$ is the tangential projection matrix; that is, $V_k(f)(x) = \sum_{j=1}^m (P_x)_{jk} \frac{\partial f}{\partial x_j}$ for any smooth function $f$. This is known as the *Riemannian Langevin diffusion* (Girolami & Calderhead, 2011). As long as $p_0$ satisfies a log-Sobolev inequality, the marginal distribution of $Y_s$ (*i.e.* the aggregated posterior) converges to $p_0$ at a linear rate in the KL divergence (Wang et al., 2020). For compact manifolds, we set $p_0$ to be the uniform density, which means $U_0 = 0$, and (14) is reduced to the extrinsic construction of Brownian motion on $\mathcal{M}$ (Hsu, 2002, Section 1.2). The benefits of this fixed-inference parameterization are the following:

**Stable and Efficient Training**. With the fixed-inference parameterization we do not need to optimize the vector fields that generate $Y_s$, and the Riemannian CT-ELBO can be rewritten as:

$$\mathbb{E}[\log p_0(Y_T)] - \int_0^T \mathbb{E}_{Y_s}\left[ \frac{1}{2}\|a(Y_s, s)\|_2^2 + \nabla_g \cdot \left( V_0 - \frac{1}{2}(V \cdot \nabla_g)V \right) \,\Bigg|\, Y_0 = x \right] \mathrm{d}s, \qquad (15)$$

where the first term is a constant wrt the model parameters (or it can be optimized separately if we want to refine the prior), and the time integral of the second term can be estimated via importance sampling (see Section 3.3). A sample of $Y_s$ can be drawn cheaply by numerically integrating (14), without requiring a stringent error tolerance (see Section 5.2 for an empirical analysis), which allows us to estimate the time integral in (15) by evaluating $a(Y_s, s)$ at a single time step $s$ only.

**Simplified Riemannian CT-ELBO**. The CT-ELBO can be simplified as the differential operator $V \cdot \nabla_g$ applied to $V$ yields a zero vector when $V$ is the tangential projection.

> **Proposition 2.** *If $V$ is the tangential projection matrix, then $(V \cdot \nabla_g)V = 0$.*

This means that we can express the generative SDE $V_0$ using the variational parameter $a$ via

$$\mathrm{d}X = (Va(X, T - t) - U_0(X, T - t))\,\mathrm{d}t + V \circ \mathrm{d}\hat{B}_t, \qquad (16)$$

with the corresponding Riemannian CT-ELBO:

$$\mathbb{E}[\log p_0(Y_T)] - \int_0^T \mathbb{E}_{Y_s} \left[ \frac{1}{2} \|a\|_2^2 + \nabla_g \cdot (Va - U_0) \,\middle|\, Y_0 = x \right] \mathrm{d}s. \tag{17}$$

## 3.3 Variance reduction

The inference process can be more generally defined to account for a time reparameterization. In fact, this leads to an equivalent model if one can find an invariant representation of the temporal variable. Learning this time rescaling can help to reduce variance (Kingma et al., 2021).

In principle, we can adopt the same methodology, but this would further complicate the parameterization of the model. Alternatively, we opt for a simpler view for variance reduction via importance sampling. We estimate the time integral "$\int \dots \mathrm{d}s$" in (17) using the following estimator:

$$\mathcal{I} := \frac{1}{q(s)} \left( \frac{1}{2} \|a\|_2^2 + \nabla_g \cdot (Va - U_0) \right) \qquad \text{where } s \sim q(s) \text{ and } Y_s \sim q(Y_s \mid Y_0), \tag{18}$$

where $q(s)$ is a proposal density supported on $[0, T]$. We parameterize $q(s)$ using a 1D monotone flow (Huang et al., 2018). As the expected value of this estimator is the same as the time integral in (17), it is unbiased. However, this means we cannot train the proposal distribution $q(s)$ by maximizing this objective, since the gradient wrt the parameters of $q(s)$ is zero in expectation. Instead, we minimize the variance of the estimator by following the stochastic gradient wrt $q(s)$

$$\nabla_{q(s)} \mathrm{Var}(\mathcal{I}) = \nabla_{q(s)} \mathbb{E}[\mathcal{I}^2] - \underbrace{\nabla_{q(s)} \mathbb{E}[\mathcal{I}]^2}_{} = \nabla_{q(s)} \mathbb{E}[\mathcal{I}^2]. \tag{19}$$

The latter can be optimized using the reparameterization trick (Kingma & Welling, 2014) and is a well-known variance reduction method in a multitude of settings (Luo et al., 2020; Tucker et al., 2017). It can be seen as minimizing the $\chi^2$-divergence from a density proportional to the magnitude of $\mathbb{E}_{Y_s}[\mathcal{I}]$ (Dieng et al., 2017; Müller et al., 2019).

## 3.4 Connection to score matching

In the Euclidean case, it can be shown that maximizing the variational lower bound of the fixed-inference diffusion model (16) is equivalent to score matching (Ho et al., 2020; Huang et al., 2021; Song et al., 2021a). In this section, we extend this connection to its Riemannian counterpart.

Let $q(y_s, s)$ be the density of $Y_s$ following (14), marginalizing out the data distribution $q(y_0, 0)$. The score function is the Riemannian gradient of the log-density $\nabla_g \log q$. The following theorem tells us that we can create a family of inference and generative SDEs that induce the same marginal distributions over $Y_s$ and $X_{T-s}$ as (16) if we have access to its score.

**Theorem 3 (Marginally equivalent SDEs).** *For $\lambda \leq 1$, the marginal distributions of $X_{T-s}$ and $Y_s$ of the processes defined as below*

$$\mathrm{d}Y = \left( U_0 - \frac{\lambda}{2} \nabla_g \log q \right) \mathrm{d}s + \sqrt{1 - \lambda} V \circ \mathrm{d}\hat{B}_s \qquad\qquad Y_0 \sim q(\cdot, 0) \tag{20}$$

$$\mathrm{d}X = \left( \left( 1 - \frac{\lambda}{2} \right) \nabla_g \log q - U_0 \right) \mathrm{d}t + \sqrt{1 - \lambda} \circ V \mathrm{d}\hat{B}_t \qquad X_0 \sim q(\cdot, T) \tag{21}$$

*both have the density $q(\cdot, s)$. In particular, $\lambda = 1$ gives rise to an equivalent ODE.*

This suggests if we can approximate the score function, and plug it into the reverse process (21), we obtain a time-reversed process that induces approximately the same marginals.

**Theorem 4 (Score matching equivalency).** *For $\lambda < 1$, let $\mathcal{E}_\lambda^\infty$ denote the Riemannian CT-ELBO of the generative process (21), with $\nabla_g \log q$ replaced by an approximate score $S_\theta$, and with (20) being the inference SDE. Assume $S_\theta$ is a compactly supported smooth vector. Then*

$$\mathbb{E}_{Y_0}[\mathcal{E}_\lambda^\infty] = -C_1 \int_0^T \mathbb{E}_{Y_s} \left[ \|S_\theta - \nabla_g \log q\|_g^2 \right] \mathrm{d}s + C_2 \tag{22}$$

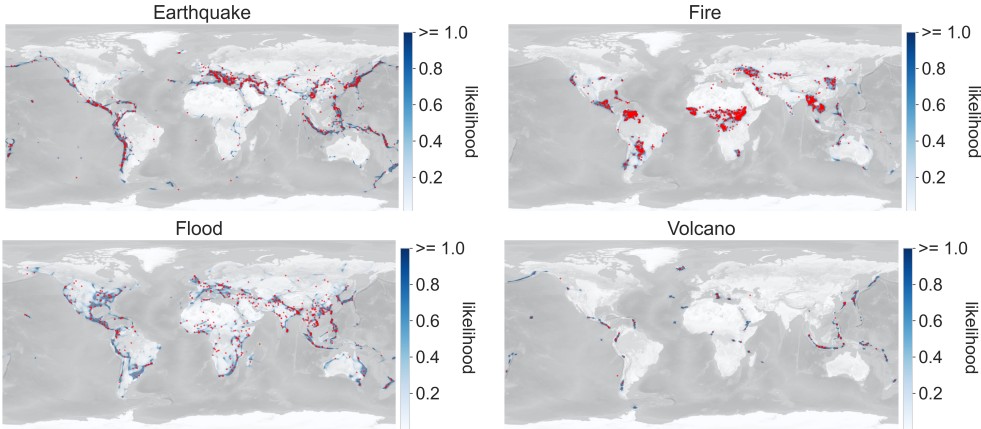

Figure 1: Density of models trained on earth datasets. Red dots are samples from the test set.

*where $C_1 > 0$ and $C_2$ are constants wrt $\theta$.*

The first implication of the theorem is that *maximizing the Riemannian CT-ELBO of the plug-in reverse process is equivalent to minimizing the Riemannian score-matching loss*. Second, if we set $\lambda = 0$, from (135) (in the appendix), we have $Va = S_\theta$, which is exactly the fixed-inference training in §3.2. That is, the vector $Va$ trained using equation (17) is actually an approximate score, allowing us to extract an equivalent ODE by substituting $Va$ for $\nabla_g \log q$ in (20,21) by setting $\lambda = 1$.

## 4 Related work

**Diffusion models**. Diffusion models can be viewed from two different but ultimately complimentary perspectives. The first approach leverages score based generative models (Song & Ermon, 2019; Song et al., 2021b), while the second approach treats generative modeling as inverting a fixed noise-injecting process (Sohl-Dickstein et al., 2015; Ho et al., 2020). Finally, continuous-time diffusion models can also be embedded within a maximum likelihood framework (Huang et al., 2021; Song et al., 2021a), which represents the special case of prescribing a flat geometry—*i.e.* Euclidean—to the generative model and is completely generalized by the theory developed in this work.

**Riemannian Generative Models**. Generative models beyond Euclidean manifolds have recently risen to prominence with early efforts focusing on constant curvature manifolds (Bose et al., 2020; Rezende et al., 2020). Another line of work extends continuous-time flows (Chen et al., 2018a) to more general Riemannian manifolds (Lou et al., 2020; Mathieu & Nickel, 2020; Falorsi & Forré, 2020). To avoid explicitly solving an ODE during training, Rozen et al. (2021) propose *Moser Flow* whose objective involves computing the Riemannian divergence of a parametrized vector field. Concurrent to our work, De Bortoli et al. (2022) develop Riemannian score-based generative models for compact manifolds like the Sphere. While similar in endeavor, RDMs are couched within the the maximum likelihood framework. As a result our approach is directly amenable to variance reduction techniques via importance sampling and likelihood estimation. Moreover, our approach is also applicable to non-compact manifolds such as hyperbolic spaces, and we demonstrate this in our experiments on a larger variety of manifolds including the orthogonal group and toroids.

## 5 Experiments

We investigate the empirical caliber of RDMs on a range of manifolds. We instantiate RDMs by parametrizing $a$ in (16) using an MLP and maximize the CT-ELBO (17). We report our detailed training procedure—including selected hyperparameters—for all models in §D.

### 5.1 Sphere

For spherical manifolds, we model the datasets compiled by Mathieu & Nickel (2020), which consist of earth and climate science events on the surface of the earth such as volcanoes (NGDC/WDS, 2022b), earthquakes (NGDC/WDS, 2022a), floods (Brakenridge, 2017), and fires (EOSDIS, 2020).

| | Volcano | Earthquake | Flood | Fire |
|---|---|---|---|---|
| Mixture of Kent | $-0.80_{\pm 0.47}$ | $0.33_{\pm 0.05}$ | $0.73_{\pm 0.07}$ | $-1.18_{\pm 0.06}$ |
| Riemannian CNF (Mathieu & Nickel, 2020) | $-0.97_{\pm 0.15}$ | $0.19_{\pm 0.04}$ | $0.90_{\pm 0.03}$ | $-0.66_{\pm 0.05}$ |
| Moser Flow (Rozen et al., 2021) | $-2.02_{\pm 0.42}$ | $-0.09_{\pm 0.02}$ | $0.62_{\pm 0.04}$ | $-1.03_{\pm 0.03}$ |
| Stereographic Score-Based | $-4.18_{\pm 0.30}$ | $-0.04_{\pm 0.11}$ | $1.31_{\pm 0.16}$ | $0.28_{\pm 0.20}$ |
| Riemannian Score-Based (De Bortoli et al., 2022) | $-5.56_{\pm 0.26}$ | $-0.21_{\pm 0.03}$ | $0.52_{\pm 0.02}$ | $-1.24_{\pm 0.07}$ |
| RDM | $\mathbf{-6.61}_{\pm 0.97}$ | $\mathbf{-0.40}_{\pm 0.05}$ | $\mathbf{0.43}_{\pm 0.07}$ | $\mathbf{-1.38}_{\pm 0.05}$ |
| Dataset size | 827 | 6120 | 4875 | 12809 |

Table 1: NLL scores for each method on earth datasets. Bold shows best results (up to statistical significance). Means and standard deviations are calculated over 5 runs. Baselines taken from De Bortoli et al. (2022).

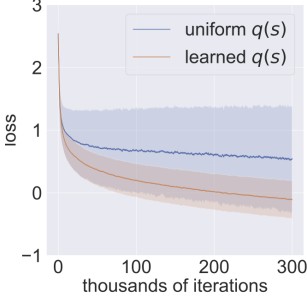

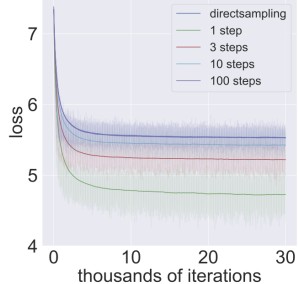

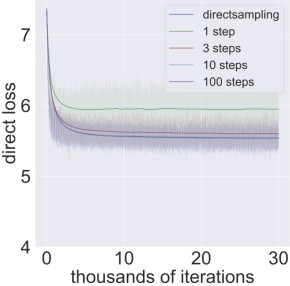

Figure 2: Variance reduction with importance sampling.

Figure 3: Direct sampling vs numerical integration of Brownian motion. Numbers in legends indicate the number of time steps.



Figure 4: Ramachandran contour plot of the model density for protein datasets. Red dots are set test samples.

In Table 1 for each dataset we report average and standard deviation of test negative log likelihood on 5 different runs with different splits of the dataset. In Figure 1 we plot the model density in blue while the test data is depicted with red dots.

**Variance reduction**. We demonstrate the effect of applying variance reduction on optimizing the Riemannian CT-ELBO (17) using the earthquake dataset. As shown in Figure 2, learning an importance sampling proposal effectively lowers the variance and speeds up training.

## 5.2 Tori

For tori, we use the list of 500 high-resolution proteins compiled in Lovell et al. (2003) and select 113 RNA sequences listed in Murray et al. (2003). Each macromolecule is divided into multiple monomers, and the joint structure is discarded—we model the lower dimensional density of the backbone conformation of the monomer. For the protein data, this corresponds to 3 torsion angles of the amino acid. As one of the angles is normally 180°, we also discard it, and model the density over the 2D torus. For the RNA data, the monomer is a nucleotide described by 7 torsion angles in the backbone, represented by a 7D torus. For protein, we divide the dataset by the type of side chain attached to the amino acid, resulting in 4 datasets, and we discard the nucleobases of the RNA.

In Table 2 we report the NLL of our model. Our baseline is a mixture of $4,096$ power spherical distributions (De Cao & Aziz, 2020, MoPS). We observe that RDM outperforms the baseline across the board, and the difference is most noticeable for the RNA data, which has a higher dimensionality.

**Numerical integration ablation**. We estimate the loss (17) by integrating the inference SDE on $\mathcal{M}$. To study the effect of integration error, we experiment with various numbers of time steps evenly spaced between $[0, s]$ on Glycine. Also, as we can directly sample the Brownian motion on tori

|  | General | Glycine | Proline | Pre-Pro | RNA |
|---|---|---|---|---|---|
| MoPS | $1.15_{\pm 0.002}$ | $2.08_{\pm 0.009}$ | $0.27_{\pm 0.008}$ | $1.34_{\pm 0.019}$ | $4.08_{\pm 0.368}$ |
| RDM | $\mathbf{1.04}_{\pm 0.012}$ | $\mathbf{1.97}_{\pm 0.012}$ | $\mathbf{0.12}_{\pm 0.011}$ | $\mathbf{1.24}_{\pm 0.004}$ | $\mathbf{-3.70}_{\pm 0.592}$ |
| Dataset size | 138208 | 13283 | 7634 | 6910 | 9478 |

Table 2: Negative test log-likelihood for each method on Tori datasets. Bold shows best results (up to statistical significance). Means and standard deviations are calculated over 5 runs.

without numerical integration, we use it as a reference (termed direct loss) for comparison. Figure 3 shows while fewer time steps tend to underestimate the loss, the model trained with 100 time steps is already indistinguishable from the one trained with direct sampling. We also find numerical integration is not a significant overhead as each experiment takes approximately the same wall-clock time with identical setups. This is because the inference path does not involve the neural module $a$.

## 5.3 Hyperbolic Manifolds

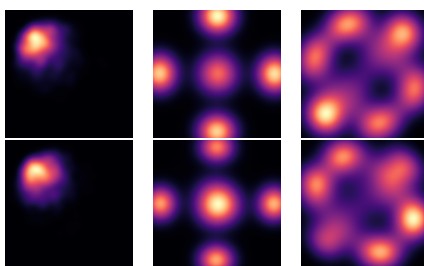

Hyperbolic manifolds provide an example whose closest-point projection is not cheap to obtain, and a claimed closest-point projection in recent literature is in fact not the closest *Euclidean projection* (Skopek et al., 2019) (see §C for more details). To demonstrate the generality of our framework, we model the synthetic datasets in Figure 5, first introduced by Bose et al. (2020); Lou et al. (2020). Since hyperbolic manifolds are not compact, we need a non-zero drift to ensure the inference processs is not dissipative. We define the prior as the standard normal

Figure 5: Hyperbolic Manifold. Top: data. Bottom: learned Density

distribution on the $yz$-plane and let $U_0$ be $\frac{1}{2}\nabla_g \log p_0$, so that $Y_s$ will revert back to the origin.

## 5.4 Special Orthogonal Group

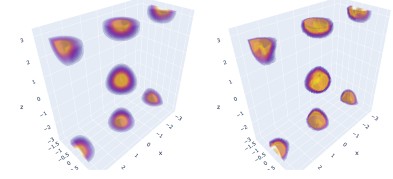

Another example whose closest-point projection is expensive to compute is the orthogonal group, as it requires performing the singular value decomposition. To evaluate our framework on this matrix group, we generate data using the synthetic multimodal density defined on $SO(3)$ from Brofos et al. (2021). We view it as a submanifold embedded in $\mathbb{R}^{3\times 3}$, therefore $d = 3$ and $m = 9$. We use the projected Hutchinson to estimate the Riemannian

Figure 6: SO(3). Left: synthetic multimodal density. Right: learned density.

divergence. Since the data are 3D rotational matrices, we can visualize them using the *Euler angles*. We plotted the data density and the learned model density in Figure 6, where each coordinate represents the rotation around that particular axis.

## 6 Conclusion

In this paper, we introduce RDMs that extend continuous-time diffusion models to arbitrary Riemannian manifolds—including challenging non-compact manifolds like hyperbolic spaces. We provide a variational framework to train RDMs by optimizing a novel objective, the Riemannian Continuous-Time ELBO. To enable efficient and stable training we provide several key tools such as a fixed-inference paramterization of the SDE in the ambient space, new methodological techniques to compute the Riemannian divergence, as well as an importance sampling procedure with respect to the time integral to reduce the variance of the loss. On a theoretical front, we also show deep connections between our proposed variational framework and Riemannian score matching through the construction of marginally equivalent SDEs. Finally, we complement our theory by constructing RDMs that achieve state-of-the-art performance on density estimation on geoscience datasets, protein/RNA data on toroidal, and synthetic data on hyperbolic and orthogonal-group manifolds.

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
