| $1.15_{\pm0.002}$ | $2.08_{\pm0.009}$ | $0.27_{\pm0.008}$ | $1.34_{\pm0.019}$ | $4.08_{\pm0.368}$ |
| RDM | $\mathbf{1.04}_{\pm0.012}$ | $\mathbf{1.97}_{\pm0.012}$ | $\mathbf{0.12}_{\pm0.011}$ | $\mathbf{1.24}_{\pm0.004}$ | $\mathbf{-3.70}_{\pm0.592}$ |
| Dataset size | 138208 | 13283 | 7634 | 6910 | 9478 |

Table 2: Negative test log-likelihood for each method on Tori datasets. Bold shows best results (up to statistical significance). Means and standard deviations are calculated over 5 runs.

without numerical integration, we use it as a reference (termed direct loss) for comparison. Figure 3 shows while fewer time steps tend to underestimate the loss, the model trained with 100 time steps is already indistinguishable from the one trained with direct sampling. We also find numerical integration is not a significant overhead as each experiment takes approximately the same wall-clock time with identical setups. This is because the inference path does not involve the neural module $a$.

## 5.3 Hyperbolic Manifolds

Hyperbolic manifolds provide an example whose closest-point projection is not cheap to obtain, and a claimed closest-point projection in recent literature is in fact not the closest *Euclidean projection* (Skopek et al., 2019) (see §C for more details). To demonstrate the generality of our framework, we model the synthetic datasets in Figure 5, first introduced by Bose et al. (2020); Lou et al. (2020). Since hyperbolic manifolds are not compact, we need a non-zero drift to ensure the inference processs is not dissipative. We define the prior as the standard normal

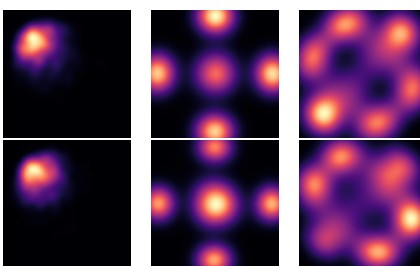

Figure 5: Hyperbolic Manifold. Top: data. Bottom: learned Density

distribution on the $yz$-plane and let $U_0$ be $\frac{1}{2}\nabla_g \log p_0$, so that $Y_s$ will revert back to the origin.

## 5.4 Special Orthogonal Group

Another example whose closest-point projection is expensive to compute is the orthogonal group, as it requires performing the singular value decomposition. To evaluate our framework on this matrix group, we generate data using the synthetic multimodal density defined on $SO(3)$ from Brofos et al. (2021). We view it as a submanifold embedded in $\mathbb{R}^{3\times3}$, therefore $d = 3$ and $m = 9$. We use the projected Hutchinson to estimate the Riemannian

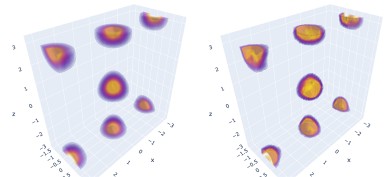

Figure 6: SO(3). Left: synthetic multimodal density. Right: learned density.

divergence. Since the data are 3D rotational matrices, we can visualize them using the *Euler angles*. We plotted the data density and the learned model density in Figure 6, where each coordinate represents the rotation around that particular axis.

## 6 Conclusion

In this paper, we introduce RDMs that extend continuous-time diffusion models to arbitrary Riemannian manifolds—including challenging non-compact manifolds like hyperbolic spaces. We provide a variational framework to train RDMs by optimizing a novel objective, the Riemannian Continuous-Time ELBO. To enable efficient and stable training we provide several key tools such as a fixed-inference paramterization of the SDE in the ambient space, new methodological techniques to compute the Riemannian divergence, as well as an importance sampling procedure with respect to the time integral to reduce the variance of the loss. On a theoretical front, we also show deep connections between our proposed variational framework and Riemannian score matching through the construction of marginally equivalent SDEs. Finally, we complement our theory by constructing RDMs that achieve state-of-the-art performance on density estimation on geoscience datasets, protein/RNA data on toroidal, and synthetic data on hyperbolic and orthogonal-group manifolds.

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

# A Riemannian manifolds

**Notation Convention.** There are a number of different notations used in differential geometry and all have their place. The most abstract level is with tensors (of which forms and vectors are special cases) and it is the best for establishing general properties. We used an index-free notation in the main paper. A coordinate-based, but still intrinsic, description that uses local charts which describe explicit coordinate systems for *patches* of the manifold: for the most part we use intrinsic coordinates in the main paper. For computational purposes it is convenient to view manifolds as hypersurfaces embedded in $\mathbb{R}^m$ even though this obscures the geometric meaning; these are called extrinsic coordinates which we use for actual implementations.

We use capital letters to denote vectors, and tilded letters to denote vectors and variables defined on the local patch.

## A.1 Smooth manifolds and tangent vectors

We recall some preliminaries of smooth manifolds. See Lee (2013) for a more detailed and comprehensive account.

A smooth $d$-manifold is a topological space $\mathcal{M}$ (assumed to be paracompact, Hausdorff and second countable) and a family of pairs $\{(U_i, \varphi_i)\}$, where the $U_i$ are open sets that together cover all of $\mathcal{M}$ and each $\varphi_i$ is a homeomorphism from $U_i$ to an open set in $\mathbb{R}^d$; these pairs are called *charts*. They are required to satisfy a compatibility condition: if $U_i$ and $U_j$ have non-empty intersection, say $V$, then $\varphi_i \circ \varphi_j^{-1}|_V$ has to be an infinitely differentiable map from $\varphi_j(V) \subset \mathbb{R}^d$ to $\varphi_i(V) \subset \mathbb{R}^d$. The use of charts allows one to talk about differentiability of functions or vectors fields, by moving to $\mathbb{R}^d$ as needed. A *smooth function* $f$ on $\mathcal{M}$ has type $\mathcal{M} \to \mathbb{R}$ and is such that for any chart $(U, \varphi)$ the map $f \circ \varphi^{-1} : \mathbb{R}^d \to \mathbb{R}$ is smooth [2]. The set of smooth functions on $\mathcal{M}$ is denoted $C^\infty(\mathcal{M})$.

Let $\mathcal{M}$ be a smooth manifold, and fix a point $x$ in $\mathcal{M}$. A **derivation** at $x$ is a linear operator $D : C^\infty(\mathcal{M}) \to \mathbb{R}$ satisfying the product rule

$$D(fg) = f(x)D(g) + g(x)D(f) \tag{23}$$

for all $f, g \in C^\infty(\mathcal{M})$. The set of all derivations at $x$ is a $d$-dimensional real vector space called the **tangent space** $\mathcal{T}_x\mathcal{M}$, and the elements of $\mathcal{T}_x\mathcal{M}$ are called the **tangent vectors** (or tangents) at $x$. For the Euclidean space $\mathcal{M} = \mathbb{R}^d$, we have that $\mathcal{T}_x\mathbb{R}^d = \text{span}\{\frac{\partial}{\partial x_1}, \cdots, \frac{\partial}{\partial x_d}\}$. We now see how to use the Euclidean derivations to induce the tangent space of arbitrary Riemannian manifolds.

Let $\mathcal{N}$ be another smooth manifold. For any tangent $V \in \mathcal{T}_x\mathcal{M}$ and smooth map $\varphi : \mathcal{M} \to \mathcal{N}$, the **differential** $d\varphi_x : \mathcal{T}_x\mathcal{M} \to \mathcal{T}_{\varphi(x)}\mathcal{N}$ is defined as the pushforward of $V$ acting on $f \in C^\infty(N)$:

$$d\varphi_x(v)(f) = V(f \circ \varphi). \tag{24}$$

Note that, if $\varphi$ is a diffeomorphism, $d\varphi_x$ is an isomorphism between $\mathcal{T}_x\mathcal{M}$ and $\mathcal{T}_{\varphi(x)}\mathcal{N}$, and the inverse map satisfies $(d\varphi_x)^{-1} = d(\varphi^{-1})_{\varphi(x)}$. Furthermore, differentials follow the chain rule, *i.e.* the differential of a composite is the composite of the differentials.

Let $\tilde{x} = (\tilde{x}_1, \cdots, \tilde{x}_d) = \varphi(x)$ be a local coordinate. Since $d\varphi_x : \mathcal{T}_x\mathcal{M} \to \mathcal{T}_{\varphi(x)}\mathbb{R}^d$ is an isomorphism, we can characterize $\mathcal{T}_x\mathcal{M}$ via inversion. We define the basis vector $\tilde{E}_i$ of $\mathcal{T}_x\mathcal{M}$ by

$$\tilde{E}_i = (d\varphi_x)^{-1}\left(\frac{\partial}{\partial \tilde{x}_i}\right) = (d\varphi^{-1})_{\varphi(x)}\left(\frac{\partial}{\partial \tilde{x}_i}\right), \tag{25}$$

which means

$$\tilde{E}_i(f) = \frac{\partial}{\partial \tilde{x}_i}f(\varphi^{-1}(\tilde{x})). \tag{26}$$

The tangent space $\mathcal{T}_x\mathcal{M}$ of $\mathcal{M}$ at $x$ is spanned by $\left\{\tilde{E}_1, \cdots, \tilde{E}_d\right\}$. This means any tangent vector $V$ can be represented by $\sum_{i=1}^d \tilde{v}_i\tilde{E}_i$ for some coordinate-dependent coefficients $\tilde{v}_i$.

---

[2]Strictly speaking this map has to be restricted to $\varphi(U)$ but we will assume that the appropriate restrictions are always intended rather than cluttering up the notation with restrictions all the time.

A manifold $\mathcal{M}$ is said to be *embedded* in $\mathbb{R}^m$ if there is an inclusion map $\iota : \mathcal{M} \to \mathbb{R}^m$ such that $\mathcal{M}$ is homeomorphic to $\iota(\mathcal{M})$ and the differential at every point is injective. Every smooth manifold can be embedded in some $\mathbb{R}^m$ with $m > d$ for some suitably chosen $m$.

When $\mathcal{M}$ is embedded in $\mathbb{R}^m$, we can view $\mathcal{T}_x\mathcal{M}$ as a linear subspace of $\mathcal{T}_x\mathbb{R}^m$; note that this map has trivial kernel. Let $\iota : \mathcal{M} \to \mathbb{R}^m$ denote the inclusion map, *i.e.* $\iota(x) = x \in \mathbb{R}^m$ for $x \in \mathcal{M}$. Then

$$\tilde{E}_i = (d\varphi^{-1})_{\varphi(x)}\left(\frac{\partial}{\partial \tilde{x}_i}\right) = (d\iota^{-1})_{\iota(x)}(d\iota \circ \varphi^{-1})_{\varphi(x)}\left(\frac{\partial}{\partial \tilde{x}_i}\right) = \sum_{j=1}^{m} \frac{\partial \varphi_j^{-1}}{\partial \tilde{x}_i}\frac{\partial}{\partial x_j}. \tag{27}$$

This means we can rewrite a tangent vector using the ambient space's basis

$$\sum_{i=1}^{d} \tilde{v}_i \tilde{E}_i = \sum_{i=1}^{d}\sum_{j=1}^{m} \tilde{v}_i \frac{\partial \varphi_j^{-1}}{\partial \tilde{x}_i}\frac{\partial}{\partial x_j} = \sum_{j=1}^{m} \bar{v}_j \frac{\partial}{\partial x_j} \tag{28}$$

where $\bar{v}_j = \sum_{i=1}^{d} \tilde{v}_i \frac{\partial \varphi_j^{-1}}{\partial \tilde{x}_i}$ is the coefficient corresponding to the $j$'th ambient space coordinate. What exactly is $\varphi_j^{-1}$? Note that $\iota \circ (\varphi^{-1})$ is a map from $\mathbb{R}^d$ to $\mathbb{R}^m$ and it takes $\varphi(x)$ to $\iota(x)$. It is this that we are writing as $\varphi_j^{-1}$.

In matrix-vector form, we can write $\bar{v} = \frac{\mathrm{d}\varphi^{-1}}{\mathrm{d}\tilde{x}}\tilde{v}$, where $\bar{v}$ is a vector that represents the $m$-dimensional coefficients in the ambient space. This also means $\bar{v}$ lies in the linear subspace spanned by the column vectors of the Jacobian $\frac{\partial \varphi^{-1}}{\partial \tilde{x}_i}$. This linear subspace is isomorphic to $\mathcal{T}_x\mathcal{M}$, which itself is a subspace of $\mathcal{T}_x\mathbb{R}^m$. We refer to this linear subspace as the **tangential linear subspace** Intuitively, this means a particle traveling at speed $\bar{v}$ and position $x$ can only move tangentially on the surface. Therefore it is restricted to move on the manifold.

A vector field $V$ is a continuous map that assigns a tangent vector to each point on the manifold; that is $V(x) \in \mathcal{T}_x\mathcal{M}$. We abuse the notation a bit and use capital letters to denote both vector fields and vectors. It should be clear in the context whether it it meant to be a function of points on the manifold or not. Such a vector field can also map a smooth function to a function, via the assignment $x \in \mathcal{M} \mapsto V(x)(f) \in \mathbb{R}$. If it maps smooth functions to smooth functions we say that the vector field is smooth. The space of smooth vector fields on $\mathcal{M}$ is denoted by by $\mathfrak{X}(\mathcal{M})$.

## A.2 Riemannian metric

A Riemannian manifold $(\mathcal{M}, g)$ is a $d$-dimensional smooth manifold $\mathcal{M}$ equipped with an inner product $g_x : \mathcal{T}_x\mathcal{M} \times \mathcal{T}_x\mathcal{M} \to \mathbb{R}$ on the tangent space of each $x \in \mathcal{M}$ (Lee, 2018). $g_x$ is called the metric tensor at $x$.

A **metric tensor field** is an assignment of a metric tensor to each point $x$ of $\mathcal{M}$; we denote it by $g$. The metric tensor field $g$ is said to be **smooth** if for any smooth vector fields $u$ and $v$, $g(U, V)(x) = g_x(U(x), V(x))$ is a smooth function of $x$. When it is clear from the context, we suppress the subscript for simplicity. Since $g$ is an inner product, we also write $g(u, v) = \langle u, v \rangle_g$.

The Euclidean metric $\bar{g}$ for $\mathbb{R}^m$ is defined as the Euclidean inner product, characterized by the delta function

$$\left\langle \frac{\partial}{\partial x_i}, \frac{\partial}{\partial x_j} \right\rangle = \delta_{ij}, \tag{29}$$

which is equal to 1 if $i = j$; otherwise it is equal to 0. This means for any $U, V \in T_x\mathcal{M}$,

$$\langle U, V \rangle_{\bar{g}} = \left\langle \sum_{i=1}^{m} \bar{u}_i \frac{\partial}{\partial x_i}, \sum_{j=1}^{m} \bar{v}_j \frac{\partial}{\partial x_j} \right\rangle_{\bar{g}} = \sum_{i=1}^{m} \bar{u}_i \bar{v}_i = \bar{u}^\top \bar{v}. \tag{30}$$

Generally, given a set of basis vectors, such as $\tilde{E}_i$, the metric tensor can be represented in a matrix form, via

$$g_{ij} := \langle \tilde{E}_i, \tilde{E}_j \rangle_g \tag{31}$$

This allows us to write the metric using the patch coordinates

$$\langle U, V \rangle_g = \sum_{i,j} \tilde{u}_i \tilde{v}_j \langle \tilde{E}_i, \tilde{E}_j \rangle_g = \sum_{i,j} \tilde{u}_i \tilde{v}_j g_{ij} = \tilde{u}^\top G \tilde{v} \tag{32}$$

where $G$ is a matrix whose $i$'th row and $j$'th column corresponds to $g_{ij}$.

Using the components of the metric tensor, we can define the **dual basis** $\tilde{E}^i = \sum_j g^{ij} \tilde{E}_j$, where $g^{ij}$ stands for the $(i, j)$'th entry of the inverse matrix $G^{-1}$. $(\tilde{E}^1, \cdots, \tilde{E}^d)$ is called the dual basis for $(\tilde{E}_1, \cdots, \tilde{E}_d)$ since they form a bi-orthogonal system, meaning

$$\langle \tilde{E}^i, \tilde{E}_j \rangle_g = \left\langle \sum_k g^{ik} \tilde{E}_k, \tilde{E}_j \right\rangle_g = \sum_k g^{ik} \langle \tilde{E}_k, \tilde{E}_j \rangle_g = \sum_k g^{ik} g_{kj} = (G^{-1} G)_{ij} = \delta_{ij}. \tag{33}$$

If $\mathcal{M}$ is a submanifold, *e.g.* if it is embedded in an ambient space, it automatically inherits the ambient manifold's metric. Suppose $\mathcal{M} \subset \mathbb{R}^m$, where $m > d$ is the dimensionality of the ambient space. Then $g = \iota^* \bar{g}$ is a metric induced by the inclusion map, defined by

$$g_x(u, v) = \bar{g}(d\iota_x(u), d\iota_x(v)).$$

Unwinding the definitions, we have

$$g_{ij} = \left\langle d\iota_x(\tilde{E}_i), d\iota_x(\tilde{E}_j) \right\rangle_{\bar{g}} = \left\langle \sum_{k=1}^m \frac{\partial \varphi_k^{-1}}{\partial \tilde{x}_i} \frac{\partial}{\partial x_k}, \sum_{k'=1}^m \frac{\partial \varphi_{k'}^{-1}}{\partial \tilde{x}_j} \frac{\partial}{\partial x_{k'}} \right\rangle_{\bar{g}} = \sum_{k=1}^m \frac{\partial \varphi_k^{-1}}{\partial \tilde{x}_i} \frac{\partial \varphi_k^{-1}}{\partial \tilde{x}_j}. \tag{34}$$

That is, if $\psi = \varphi^{-1}$ is the inverse map of $\varphi$, we can write $G = \frac{d\psi}{d\tilde{x}}^\top \frac{d\psi}{d\tilde{x}}$, which can be equivalently deduced from equating (30) and (32).

An important use of the metric is to define a measure over measurable subsets of the manifold. Let $(U, \varphi)$ be a chart and consider all functions smooth functions $f$ supported in $U$. Then

$$f \mapsto \int_{\varphi(U)} (f \sqrt{|\det G|}) \circ \varphi^{-1} \, d\tilde{x}$$

is a positive linear functional. Since $\mathcal{M}$ is Hausdorff and locally compact, by the *Riesz representation theorem* (Rudin, 1987, Theorem 2.14), there exists a unique Borel measure $\mu_g$ (over $U$) such that $\int_U f \, d\mu_g$ is equal to the evaluation of the functional above. We can then apply a partition of unity (Lee, 2013, Theorem 2.23) to extend this construction of $\mu_g$ to be defined over the entire $\mathcal{M}$, which says that for any open cover $\{U_i\}$ of $\mathcal{M}$, there exists a set of continuous functions $\Phi_i$ satisfying the following properties:

1. $0 \leq \Phi_i(x)$ for all $x \in \mathcal{M}$.

2. $\operatorname{supp} \Phi_i \subseteq U_i$.

3. $\sum_i \Phi_i(x) = 1$ for all $x \in \mathcal{M}$.

4. Any $x \in \mathcal{M}$ has a neighborhood that intersects with only finitely many $\operatorname{supp} \Phi_i$.

By means of the partition, we can consider the following positive linear functional instead:

$$f \in C_c(\mathcal{M}) \mapsto \sum_i \int_{\varphi(U_i)} (\Psi_i f \sqrt{|\det G|}) \circ \varphi^{-1} \, d\tilde{x}, \tag{35}$$

which is always well-defined since $f$ is compactly supported in $\mathcal{M}$ (only finitely many summands are non-zero). $\sqrt{|\det G|}$ is called the **volume density**. We write $|G| = |\det G|$ for short. A probability density $p$ over $\mathcal{M}$ can be thought of as a non-negative integrable function satisfying $\int_{\mathcal{M}} p \, d\mu_g = 1$.

## A.3 Riemannian gradient and divergence

**Riemannian gradient**  Another crucial structure closely related to the metric is the **Riemannian gradient**. The definition of Riemannian gradient $\nabla_g : f \in C^\infty(\mathcal{M}) \mapsto \nabla_g f \in \mathfrak{X}(\mathcal{M})$ is motivated by the directional derivative in Euclidean space, satisfying

$$\langle \nabla_g f, V \rangle_g = V(f) \tag{36}$$

for any $V \in \mathfrak{X}(\mathcal{M})$.

To obtain an explicit formula for the Riemannian gradient, we expand both sides of (36):

$$\langle \nabla_g f, V \rangle_g = \sum_{i,j=1}^{d} \tilde{u}_i \tilde{v}_j g_{ij} \tag{37}$$

where we let $\tilde{u}_i$ and $\tilde{v}_j$ denote the coefficients of the gradient and $V$ respectively. And,

$$V(f) = \sum_{j=1}^{d} \tilde{v}_j \frac{\partial}{\partial \tilde{x}_j} f \circ \varphi^{-1}. \tag{38}$$

Since $v$ is arbitrary, this means for all $j$

$$\sum_{i=1}^{d} \tilde{u}_i g_{ij} = \frac{\partial}{\partial \tilde{x}_j} f \circ \varphi^{-1} \quad \Longrightarrow \quad \tilde{u}_i = \sum_{j=1}^{d} g^{ij} \frac{\partial}{\partial \tilde{x}_j} f \circ \varphi^{-1}. \tag{39}$$

**Riemannian divergence**  Recall that we define the Riemannian divergence using the patch coordinates in (6), which we later show has a coordinate-free form (11) and can be computed in the ambient space (12) if the manifold is embedded. The following theorem extends the Stokes theorem to Riemannian manifolds.

> **Theorem 5 (Divergence theorem).** *For any compactly supported $f \in \mathfrak{X}(\mathcal{M})$, $\int_{\mathcal{M}} \nabla_g \cdot f \, \mathrm{d}\mu_g = 0$.*

*Proof.*  Let $\{(\Psi_i, U_i)\}$ be a partition of unity. By compactness, we can choose a finite subcover over the support of $f$, so the index set of $i$ is finite.

$$\int_{\mathcal{M}} \nabla_g \cdot f \, \mathrm{d}\mu_g = \int_{\mathcal{M}} \nabla_g \cdot \left( \sum_i \Psi_i f \right) \mathrm{d}\mu_g \tag{40}$$

$$= \sum_i \int_{U_i} \nabla_g \cdot (\Psi_i f) \, \mathrm{d}\mu_g \tag{41}$$

$$= \sum_i \int_{\varphi_i(U_i)} \nabla \cdot (|G|^{\frac{1}{2}} \Psi_i f) \circ \varphi^{-1} \, \mathrm{d}\tilde{x}. \tag{42}$$

All of the finitely many summands equal 0 by an application of Stokes' theorem in $\mathbb{R}^d$ (Rudin et al., 1976, Theorem 10.33). This is because the support of $\Psi_i \circ \varphi_i^{-1}$ is contained in $\varphi_i(U_i)$); therefore at the boundary of $\varphi_i(U_i)$, $\Psi_i \circ \varphi_i^{-1}$ is equal to 0.

$\square$

The Riemannian divergence satisfies the following product rule.

> **Proposition 3 (Product rule).** *Assume $V \in \mathfrak{X}(\mathcal{M})$ and $f \in C^\infty(\mathcal{M})$. Then*
> $$\nabla_g \cdot (fV) = V(f) + f \nabla_g \cdot V. \tag{43}$$

*Proof.* Using (11), the product rule of the Affine connection (see Appendix A.4),

$$\nabla_g \cdot (fV) = \sum_{j=1}^{d} \langle \nabla_{\tilde{E}_j}(fV), \tilde{E}^j \rangle_g \tag{44}$$

$$= \sum_{j=1}^{d} \langle f \nabla_{\tilde{E}_j} V + \tilde{E}_j(f)V, \tilde{E}^j \rangle_g \tag{45}$$

$$= f \sum_{j=1}^{d} \langle \nabla_{\tilde{E}_j} V, \tilde{E}^j \rangle_g + \sum_{j=1}^{d} \tilde{E}_j(f) \left\langle \sum_{j'=1}^{d} \tilde{v}_{j'} \tilde{E}_{j'}, \tilde{E}^j \right\rangle_g \tag{46}$$

$$= f \nabla_g \cdot V + \sum_{j,j'=1}^{d} \tilde{E}_j(f) \tilde{v}_{j'} \langle \tilde{E}_{j'}, \tilde{E}^j \rangle_g \tag{47}$$

$$= f \nabla_g \cdot V + \sum_{j,j'=1}^{d} \tilde{E}_j(f) \tilde{v}_{j'} \delta_{jj'} \tag{48}$$

$$= f \nabla_g \cdot V + \sum_{j}^{d} \tilde{E}_j(f) \tilde{v}_j = f \nabla_g \cdot V + V(f). \tag{49}$$

$\square$

> **Proposition 4 (Expanding Riemannian gradient).** *Let $V$ denote the tangential projection matrix in the sense of Proposition 2. Then for any $f \in C^\infty(\mathcal{M})$*
>
> $$\sum_{k=1}^{d} V_k(f) V_k = \nabla_g f. \tag{50}$$

### A.4   Covariant derivative

An **affine connection** allows us to compare values of a vector field at nearby points. It is a differential operator denoted by $\nabla : \mathfrak{X}(\mathcal{M}) \times \mathfrak{X}(\mathcal{M}) \to \mathfrak{X}(\mathcal{M})$ and written as $U, V \mapsto \nabla_U V$ for $U, V \in \mathfrak{X}(\mathcal{M})$, satisfying the following defining properties:

1. Linearity in $U$: $\nabla_{fU_1 + gU_2} V = f \nabla_{U_1} V + g \nabla_{U_2} V$ for $f, g \in C^\infty(M)$ and $U_1, U_2, V \in \mathfrak{X}(M)$.
2. Linearity in $V$: $\nabla_U(aV_1 + bV_2) = a \nabla_U V_1 + b \nabla_U V_2$ for $a, b \in \mathbb{R}$ and $U, V_1, V_2 \in \mathfrak{X}(M)$.
3. Product rule: $\nabla_U(fV) = f \nabla_U V + U(f)V$ for $f \in C^\infty(M)$ and $U, V \in \mathfrak{X}(M)$.

$\nabla_U V$ is called the **covariant derivative** of $V$ in the $U$-direction.

If $U, V \in \mathfrak{X}(\mathbb{R}^m)$, the Euclidean connection $\overline{\nabla}$ is defined as

$$\overline{\nabla}_U V = \sum_{i=1}^{m} \sum_{j=1}^{m} \bar{u}_j \frac{\partial \bar{v}_i}{\partial x_j} \frac{\partial}{\partial x_i}. \tag{51}$$

It can be verified that the Euclidean connection is indeed an affine connection.

We can express a connection internally in terms of a coordinate system $\tilde{E}_i$. For any pair of indices $i$ and $j$, we define the connection coefficients of $\nabla$, denoted by $\Gamma$, as $d^3$ smooth functions satisfying

$$\nabla_{\tilde{E}_i} \tilde{E}_j = \sum_{k=1}^{d} \Gamma_{ij}^k \tilde{E}_k. \tag{52}$$

Then for any $U, V \in \mathfrak{X}(\mathcal{M})$, we have

$$\nabla_U V = \nabla_U \sum_{j=1}^{d} \tilde{v}_j \tilde{E}_j \tag{53}$$

$$= \sum_{j=1}^{d} \tilde{v}_j \nabla_U \tilde{E}_j + U(\tilde{v}_j) \tilde{E}_j \tag{54}$$

$$= \sum_{i,j=1}^{d} \tilde{u}_i \tilde{v}_j \nabla_{\tilde{E}_i} \tilde{E}_j + \sum_{j=1}^{d} U(\tilde{v}_j) \tilde{E}_j \tag{55}$$

$$= \sum_{i,j,k=1}^{d} \tilde{u}_i \tilde{v}_j \Gamma_{ij}^k \tilde{E}_k + \sum_{j=1}^{d} U(\tilde{v}_j) \tilde{E}_j. \tag{56}$$

Now given a metric tensor, we say that $\nabla$ is a **Levi-Civita connection of** $g$ if it is

1. Compatible with $g$: $U(g(V, W)) = g(\nabla_U V, W) + g(V, \nabla_U W)$.
2. Symmetric: $\nabla_u v - \nabla_V U = [U, V]$, where $[U, V] := \sum_{i=1}^{d} U(V_i) \tilde{E}_i - V(U_i) \tilde{E}_i$ is the Lie bracket.

The first condition looks messy but it essentially says that the Levi-Civita connection leaves the metric invariant. It is equivalent to saying that the covariant derivative of $g$ in any direction is zero.

> **Theorem 6** (**Fundamental Theorem of Riemannian Geometry**). *Let $(\mathcal{M}, g)$ be a Riemannian manifold. There exists a unique Levi-Civita connection of $g$.*

See Lee (2018, Theorem 5.10) for proof. The connection coefficients of the Levi-Civita connection are called the **Christoffel symbols** of $g$. They are symmetric in the lower indices, *i.e.* $\Gamma_{ij}^k = \Gamma_{ji}^k$. A by-product of the proof of the fundamental theorem is the following identity, which will turn out to be useful in deriving the identity for the Riemannian divergence:

$$\frac{\partial}{\partial \tilde{x}_j} g_{ki} = \sum_{l=1}^{d} \Gamma_{jk}^l g_{li} + \Gamma_{ji}^l g_{lk}. \tag{57}$$

An example of a Levi-Civita connection is the Euclidean connection of $(\mathbb{R}^d, \bar{g})$. It can be checked that $\overline{\nabla}$ is both symmetric and compatible with $\bar{g}$. Furthermore, for any $d$-submanifold $\mathcal{M}$ embedded in $\mathbb{R}^m$ for $m > d$, we can define a **tangential connection**

$$\nabla_U^\top V = P \overline{\nabla}_{\overline{U}} \overline{V} \tag{58}$$

for $U, V \in \mathfrak{X}(\mathcal{M})$, where $\overline{U}$ and $\overline{V}$ are any[3] smooth extensions of $U$ and $V$ to $\mathbb{R}^m$. $P$ is the tangential projection defined as

$$(PV)(x) = \sum_{j=1}^{m} (P_x \bar{v})_j \frac{\partial}{\partial x_j} \tag{59}$$

for any $V \in \mathfrak{X}(\mathbb{R}^m)$. Recall that $P_x$ is the **orthogonal projection** onto the tangent space spanned by $\frac{\partial \psi}{\partial \tilde{x}_i}$. *The tangential connection $\nabla^\top$ is the Levi-Civita connection on the embedded submanifold $\mathcal{M}$* (Lee, 2018, Proposition 5.12).

---

[3]The value of the tangential connection is independent of the extensions chosen, so $\nabla^\top$ is well-defined.

## B Proofs

**Theorem 1** (**Marginal Density**). *The density $p(x,t)$ of the SDE (5) can be written as*

$$p(x,t) = \mathbb{E}\left[p_0\left(Y_t\right)\exp\left(-\int_0^t \nabla_g \cdot \left(V_0 - \frac{1}{2}(V\cdot\nabla_g)V\right)ds\right) \;\middle|\; Y_0 = x\right] \qquad (7)$$

*where the expectation is taken wrt the following process induced by a Brownian motion $B_s'$*

$$dY = \left(-V_0 + (V\cdot\nabla_g)V\right)ds + V\circ dB_s'. \qquad (8)$$

*Proof.* Our first step is to express the time derivative of the density using *derivations* (spatial derivatives); this gives us a partial differential equation (PDE) on the manifold. Second, we apply the Feynman-Kac formula (Thalmaier, 2021, Proposition 3.1) to the solution of the PDE.

We denote by $d\tilde{X}_t = \tilde{v}_0\,dt + \tilde{v}\circ dB_t$ the Stratonovich SDE defined on the patch. The density $p$ of the process satisfies the Fokker-Planck equation (Chirikjian, 2009, Equation (8.16)):

$$\partial_t p(\tilde{x},t) = \underbrace{-|G|^{-\frac{1}{2}}\nabla\cdot(|G|^{\frac{1}{2}}\tilde{v}_0 p)}_{\text{first term}} + \underbrace{\frac{1}{2}|G|^{-\frac{1}{2}}\sum_{i=1}^d\sum_{j=1}^d\frac{\partial}{\partial\tilde{x}_i}\left(\sum_{k=1}^w \tilde{v}_{i,k}\frac{\partial}{\partial\tilde{x}_j}\left(|G|^{\frac{1}{2}}\tilde{v}_{j,k}p\right)\right)}_{\text{second term}} \qquad (60)$$

We would like to re-express the RHS using the abstract vectors $V_0$ and $V$. Note the first term can be written as $-\nabla_g\cdot(pV_0)$. We now show that we can also rewrite the second term in terms of the Riemannian divergence.

$$\frac{1}{2}|G|^{-\frac{1}{2}}\sum_{i=1}^d\sum_{j=1}^d\frac{\partial}{\partial\tilde{x}_i}\left(\sum_{k=1}^w \tilde{v}_{i,k}\frac{\partial}{\partial\tilde{x}_j}\left(|G|^{\frac{1}{2}}\tilde{v}_{j,k}p\right)\right) \qquad (61)$$

$$= \frac{1}{2}|G|^{-\frac{1}{2}}\sum_{k=1}^w\sum_{i=1}^d\frac{\partial}{\partial\tilde{x}_i}\left(\tilde{v}_{i,k}\sum_{j=1}^d\frac{\partial}{\partial\tilde{x}_j}\left(|G|^{\frac{1}{2}}\tilde{v}_{j,k}p\right)\right) \qquad (62)$$

$$= \frac{1}{2}|G|^{-\frac{1}{2}}\sum_{k=1}^w\sum_{i=1}^d\frac{\partial}{\partial\tilde{x}_i}\left(\tilde{v}_{i,k}|G|^{\frac{1}{2}}|G|^{-\frac{1}{2}}\sum_{j=1}^d\frac{\partial}{\partial\tilde{x}_j}\left(|G|^{\frac{1}{2}}\tilde{v}_{j,k}p\right)\right) \qquad (63)$$

$$= \frac{1}{2}|G|^{-\frac{1}{2}}\sum_{k=1}^w\sum_{i=1}^d\frac{\partial}{\partial\tilde{x}_i}\left(\tilde{v}_{i,k}|G|^{\frac{1}{2}}\nabla_g\cdot(pV_k)\right) \qquad (64)$$

$$= \frac{1}{2}\sum_{k=1}^w|G|^{-\frac{1}{2}}\sum_{i=1}^d\frac{\partial}{\partial\tilde{x}_i}\left(|G|^{\frac{1}{2}}\tilde{v}_{i,k}\nabla_g\cdot(pV_k)\right) \qquad (65)$$

$$= \frac{1}{2}\sum_{k=1}^w\nabla_g\cdot\left(\left(\nabla_g\cdot(pV_k)\right)V_k\right) \qquad (66)$$

Summing these two terms give us

$$\partial_t p(x,t) = -\nabla_g\cdot(pV_0) + \frac{1}{2}\sum_{k=1}^w\nabla_g\cdot\left(\left(\nabla_g\cdot(pV_k)\right)V_k\right) \qquad (67)$$

Next, we expand the above formula using the product rule (43):

$$\partial_t p(x,t) = -\nabla_g \cdot (pV_0) + \frac{1}{2}\sum_{k=1}^{w} \nabla_g \cdot \left(\left(\nabla_g \cdot (pV_k)\right)V_k\right) \tag{68}$$

$$= -V_0(p) - p\nabla_g \cdot (V_0) + \frac{1}{2}\sum_{k=1}^{w} \nabla_g \cdot \left(\left(V_k(p) + p\nabla_g \cdot (V_k)\right)V_k\right) \tag{69}$$

$$= -V_0(p) - p\nabla_g \cdot (V_0) + \frac{1}{2}\sum_{k=1}^{w} \nabla_g \cdot \left(\left(V_k(p) + p\nabla_g \cdot (V_k)\right)V_k\right) \tag{70}$$

$$= -V_0(p) - p\nabla_g \cdot (V_0) + \frac{1}{2}\sum_{k=1}^{w} \left(V_k\left(V_k(p) + p\nabla_g \cdot (V_k)\right) + \left(V_k(p) + p\nabla_g \cdot (V_k)\right)\nabla_g \cdot (V_k)\right) \tag{71}$$

$$= -V_0(p) - p\nabla_g \cdot (V_0) + \frac{1}{2}\sum_{k=1}^{w} \left(V_k(V_k(p)) + V_k(p\nabla_g \cdot (V_k)) + V_k(p)\nabla_g \cdot (V_k) + p\nabla_g \cdot (V_k)\nabla_g \cdot (V_k)\right) \tag{72}$$

$$= -V_0(p) - p\nabla_g \cdot (V_0) + \frac{1}{2}\sum_{k=1}^{w} \left(V_k(V_k(p)) + V_k(p\nabla_g \cdot (V_k)) + V_k(p)\nabla_g \cdot (V_k) + p(\nabla_g \cdot (V_k))^2\right) \tag{73}$$

$$= -V_0(p) - p\nabla_g \cdot (V_0) + \frac{1}{2}\sum_{k=1}^{w} \left(V_k^2(p) + V_k(p\nabla_g \cdot (V_k)) + V_k(p)\nabla_g \cdot (V_k) + p(\nabla_g \cdot (V_k))^2\right) \tag{74}$$

$$= -V_0(p) - p\nabla_g \cdot (V_0) + \frac{1}{2}\sum_{k=1}^{w} \left(V_k^2(p) + V_k(p)\nabla_g \cdot (V_k) + pV_k(\nabla_g \cdot (V_k)) + V_k(p)\nabla_g \cdot (V_k) + p(\nabla_g \cdot (V_k))^2\right) \tag{75}$$

$$= -V_0(p) - p\nabla_g \cdot (V_0) + \frac{1}{2}\sum_{k=1}^{w} \left(V_k^2(p) + V_k(p)\nabla_g \cdot (V_k) + pV_k(\nabla_g \cdot (V_k)) + V_k(p)\nabla_g \cdot (V_k) + p(\nabla_g \cdot (V_k))^2\right) \tag{76}$$

$$= -V_0(p) - p\nabla_g \cdot (V_0) + \sum_{k=1}^{w} V_k(p)\nabla_g \cdot (V_k) + \frac{1}{2}\sum_{k=1}^{w} \left(V_k^2(p) + pV_k(\nabla_g \cdot (V_k)) + p(\nabla_g \cdot (V_k))^2\right) \tag{77}$$

$$= -V_0(p) - p\nabla_g \cdot (V_0) + \sum_{k=1}^{w} \left(V_k\nabla_g \cdot (V_k)\right)(p) + \frac{1}{2}\sum_{k=1}^{w} \left(V_k^2(p) + pV_k(\nabla_g \cdot (V_k)) + p(\nabla_g \cdot (V_k))^2\right) \tag{78}$$

$$= -V_0(p) - p\nabla_g \cdot (V_0) + \left((V \cdot \nabla_g)V\right)(p) + \frac{1}{2}\sum_{k=1}^{w} \left(V_k^2(p) + pV_k(\nabla_g \cdot (V_k)) + p(\nabla_g \cdot (V_k))^2\right) \tag{79}$$

$$= -V_0(p) - p\nabla_g \cdot (V_0) + \left((V \cdot \nabla_g)V\right)(p) + \frac{1}{2}\sum_{k=1}^{w} \left(V_k^2(p) + p\nabla_g \cdot ((\nabla_g \cdot V_k)V_k)\right) \tag{80}$$

$$= -V_0(p) - p\nabla_g \cdot (V_0) + \left((V \cdot \nabla_g)V\right)(p) + \frac{1}{2}\sum_{k=1}^{w} \left(V_k^2(p) + p\nabla_g \cdot ((\nabla_g \cdot V_k)V_k)\right) \tag{81}$$

$$= -V_0(p) - p\nabla_g \cdot (V_0) + \left((V \cdot \nabla_g)V\right)(p) + \frac{1}{2}\sum_{k=1}^{w} V_k^2(p) + \frac{1}{2}\sum_{k=1}^{w} p\nabla_g \cdot ((\nabla_g \cdot V_k)V_k) \tag{82}$$

$$= -V_0(p) - p\nabla_g \cdot (V_0) + \left((V \cdot \nabla_g)V\right)(p) + \frac{1}{2}\sum_{k=1}^{w} V_k^2(p) + \frac{1}{2}p\nabla_g \cdot ((V \cdot \nabla_g)V) \tag{83}$$

In order to apply the Feynman-Kac formula, we group all the terms by the order of differentiation (of $p$), which gives us

$$\partial_t p(x,t) = -V_0(p) - p\nabla_g \cdot (V_0) + \left((V \cdot \nabla_g)V\right)(p) + \frac{1}{2}\sum_{k=1}^{w} V_k^2(p) + \frac{1}{2}p\nabla_g \cdot ((V \cdot \nabla_g)V)$$

(84)

$$= p\underbrace{\left(-\nabla_g \cdot (V_0) + \frac{1}{2}\nabla_g \cdot ((V \cdot \nabla_g)V))\right)}_{\mathcal{V}} + \left(-V_0 + ((V \cdot \nabla_g)V)\right)(p) + \left(\frac{1}{2}\sum_{k=1}^{w} V_k^2\right)(p)$$

(85)

Now the above is a parabolic PDE, which can be solved using the Feynman-Kac formula (Thalmaier, 2021, Proposition 3.1). Let $Y$ be induced (8) restated below

$$\begin{cases} \mathrm{d}Y = (-V_0 + (V \cdot \nabla_g)V)dt + \sum_{k=1}^{w}(V_k) \circ \mathrm{d}B'^{k}_{s} \\ Y_0 = x \end{cases}$$

(86)

Then $p(x,t)$ is given by

$$p(x,t) = \mathbb{E}\left[\exp\left(\int_0^t \mathcal{V}\left(Y_s(x)\right)ds\right)p_0\left(Y_t\right) \Big| Y_0 = x\right]$$

(87)

where $p_0 = p(x,0)$ is the prior distribution.

$\square$

**Theorem 2** (**Riemannian CT-ELBO**). *Let $\hat{B}_s$ be a $w$-dimensional Brownian motion, and let $Y_s$ be a process solving the following*

$$\text{Inference SDE:} \qquad \mathrm{d}Y = (-V_0 + (V \cdot \nabla_g)V + Va) \, \mathrm{d}s + V \circ \mathrm{d}\hat{B}_s, \qquad (9)$$

*where $a : \mathbb{R}^m \times [0, T] \to \mathbb{R}^m$ is the variational degree of freedom. Then we have*

$$\log p(x, T) \geq \mathbb{E} \left[ \log p_0(Y_T) - \int_0^T \frac{1}{2} \|a(Y_s, s)\|_2^2 + \nabla_g \cdot \left( V_0 - \frac{1}{2}(V \cdot \nabla_g)V \right) \mathrm{d}s \,\middle|\, Y_0 = x \right],$$
$$(10)$$

*where all the generative degree of freedoms $V_k$ are evaluated in the reversed time direction.*

*Proof.* Let $\mathbb{P}$ be the probability measure under which $B'$ is a Brownian motion. Let

$$\mathrm{d}\hat{B} = -a \, \mathrm{d}s + \mathrm{d}B'_s, \qquad (88)$$

where $a$ is the variational degree of freedom. Let $\mathbb{Q}$ be defined as

$$\mathrm{d}\mathbb{Q} = \exp \left( \int_0^T a(Y_s, s)\mathrm{d}B'_s - \frac{1}{2} \int_0^T \|a(Y_s, s)\|_2^2 \, \mathrm{d}s \right) \mathrm{d}\mathbb{P}. \qquad (89)$$

Note that the first term is an Itô integral. Then by the Girsanov theorem (Øksendal, 2003, Theorem 8.6.3), $\hat{B}$ is a Brownian motion wrt $\mathbb{Q}$. Therefore, changing the measure from $\mathbb{P}$ to $\mathbb{Q}$ to the expression in Theorem 1 yields

$$\log p(x, t) = \log \mathbb{E}_{\mathbb{Q}} \left[ \frac{\mathrm{d}\mathbb{P}}{\mathrm{d}\mathbb{Q}} \cdot p_0 (Y_t) \exp \left( -\int_0^T \nabla_g \cdot \left( V_0 - \frac{1}{2}(V \cdot \nabla_g)V \right) ds \right) \,\middle|\, Y_0 = x \right],$$

which by Jensen's inequality, is lower bounded by

$$\mathbb{E}_{\mathbb{Q}} \left[ \log \frac{\mathrm{d}\mathbb{P}}{\mathrm{d}\mathbb{Q}} + \log p_0 (Y_t) - \left( \int_0^T \nabla_g \cdot \left( V_0 + \frac{1}{2}(V \cdot \nabla_g)V \right) ds \right) \,\middle|\, Y_0 = x \right]. \qquad (90)$$

Now under the expectation, the Radon-Nikodym derivative can be simplified:

$$\mathbb{E}_{\mathbb{Q}} \left[ \log \frac{\mathrm{d}\mathbb{P}}{\mathrm{d}\mathbb{Q}} \,\middle|\, Y_0 = x \right] = \mathbb{E}_{\mathbb{Q}} \left[ -\int_0^T a(Y_s, s)\mathrm{d}B'_s + \frac{1}{2} \int_0^T \|a(Y_s, s)\|_2^2 \, \mathrm{d}s \,\middle|\, Y_0 = x \right] \qquad (91)$$

$$= \mathbb{E}_{\mathbb{Q}} \left[ -\int_0^T a(Y_s, s)\mathrm{d}\hat{B}_s - \frac{1}{2} \int_0^T \|a(Y_s, s)\|_2^2 \, \mathrm{d}s \,\middle|\, Y_0 = x \right] \qquad (92)$$

$$= \mathbb{E}_{\mathbb{Q}} \left[ -\frac{1}{2} \int_0^T \|a(Y_s, s)\|_2^2 \, \mathrm{d}s \,\middle|\, Y_0 = x \right] \qquad (93)$$

where we used the definition of $\mathbb{Q}$ (89), the definition of $\mathrm{d}\hat{B}$ (88), and the Martingale property of the Itô integral (Øksendal, 2003, Corollary 3.2.6). This concludes the proof.

$\square$

**Proposition 1** (**Riemannian divergence identity**). *Let $(M, g)$ be a $d$-dimensional Rieman-nian manifold. For any smooth vector field $V_k \in \mathfrak{X}(\mathcal{M})$, the following identity holds:*

$$\nabla_g \cdot V_k = \sum_{j=1}^{d} \left\langle \nabla_{\tilde{E}_j} V_k, \tilde{E}^j \right\rangle_g. \tag{11}$$

*Furthermore, if the manifold is a submanifold embedded in the ambient space $\mathbb{R}^m$ equipped with the induced metric $g = \iota^* \bar{g}$, then*

$$(\nabla_g \cdot V_k)(x) = \operatorname{tr}\left( P_x \frac{\mathrm{d} v_k}{\mathrm{d} x} P_x \right), \tag{12}$$

*where $v_k = (v_{1k}, \cdots, v_{mk})$ are the ambient space coefficients $V_k = \sum_{i=1}^{m} v_{ik} \frac{\partial}{\partial x_i}$ and $P_x$ is the orthogonal projection onto the tangent space.*

*Proof.* We drop the index on $k$ (since the statement is for any smooth vector). Using product rule, the LHS of (11) is equal to

$$\sum_{j=1}^{d} \frac{\partial \tilde{v}_j}{\partial \tilde{x}_j} + \tilde{v}_j |G|^{-\frac{1}{2}} \frac{\partial}{\partial \tilde{x}_j} |G|^{\frac{1}{2}} \tag{94}$$

Using the chain rule, Jacobi's formula, and the identity (57), we have

$$\tilde{v}_j |G|^{-\frac{1}{2}} \frac{\partial}{\partial \tilde{x}_j} |G|^{\frac{1}{2}} = \frac{1}{2} \tilde{v}_j |G|^{-1} \frac{\partial}{\partial \tilde{x}_j} \det G \tag{95}$$

$$= \frac{1}{2} \tilde{v}_j \operatorname{tr}\left( G^{-1} \frac{\partial G}{\partial \tilde{x}_j} \right) \tag{96}$$

$$= \frac{1}{2} \tilde{v}_j \sum_{i,k=1}^{d} g^{ik} \frac{\partial g_{ki}}{\partial \tilde{x}_j} \tag{97}$$

$$= \frac{1}{2} \tilde{v}_j \sum_{i,k=1}^{d} g^{ik} \left( \sum_{l=1}^{d} \Gamma^l_{jk} g_{li} + \Gamma^l_{ji} g_{lk} \right) \tag{98}$$

$$= \frac{1}{2} \tilde{v}_j \left( \sum_{i,k,l=1}^{d} \Gamma^l_{jk} g^{ik} g_{li} + \tilde{v}_j \sum_{i,k,l=1}^{d} \Gamma^l_{ji} g^{ik} g_{lk} \right) \tag{99}$$

$$= \frac{1}{2} \tilde{v}_j \sum_{k,l=1}^{d} \Gamma^l_{jk} \delta_{kl} + \frac{1}{2} \tilde{v}_j \sum_{i,l=1}^{d} \Gamma^l_{ji} \delta il \tag{100}$$

$$= \frac{1}{2} \tilde{v}_j \sum_{k=1}^{d} \Gamma^k_{jk} + \frac{1}{2} \tilde{v}_j \sum_{i=1}^{d} \Gamma^i_{ji} \tag{101}$$

$$= \tilde{v}_j \sum_{k=1}^{d} \Gamma^k_{jk} \tag{102}$$

Therefore, the LHS reduces to

$$\sum_{j=1}^{d} \left( \frac{\partial \tilde{v}_j}{\partial \tilde{x}_j} + \tilde{v}_j \sum_{k=1}^{d} \Gamma^k_{jk} \right) \tag{103}$$

Now we express the covariant derivative on the RHS using the connection coefficients (56)

$$\nabla_{\tilde{E}_j} V = \sum_{i,k=1}^{d} \tilde{v}_i \Gamma^k_{ji} \tilde{E}_k + \sum_{i=1}^{d} \frac{\partial \tilde{v}_i}{\partial \tilde{x}_j} \tilde{E}_i \tag{104}$$

which means

$$\langle \nabla_{\tilde{E}_j} V, \tilde{E}^j \rangle_g = \sum_{i,k=1}^d \tilde{v}_i \Gamma_{ji}^k \langle \tilde{E}_k, \tilde{E}^j \rangle_g + \sum_{i=1}^d \frac{\partial \tilde{v}_i}{\partial \tilde{x}_j} \langle \tilde{E}_i, \tilde{E}^j \rangle_g \tag{105}$$

$$= \sum_{i,k=1}^d \tilde{v}_i \Gamma_{ji}^k \delta_{kj} + \sum_{i=1}^d \frac{\partial \tilde{v}_i}{\partial \tilde{x}_j} \delta_{ij} \tag{106}$$

$$= \sum_{i=1}^d \tilde{v}_i \Gamma_{ji}^j + \frac{\partial \tilde{v}_j}{\partial \tilde{x}_j}. \tag{107}$$

Now summing all terms yields

$$\sum_{j=1}^d \langle \nabla_{\tilde{E}_j} V, \tilde{E}^j \rangle_g = \sum_{i,j=1}^d \tilde{v}_i \Gamma_{ji}^j + \sum_{j=1}^d \frac{\partial \tilde{v}_j}{\partial \tilde{x}_j}. \tag{108}$$

Relabeling $i \to j$ and $j \to k$ in the term term shows this is equal to the LHS.

For the second half of the theorem, recall that the Levi-Civita connection is equal to the tangential connection. Therefore, changing the basis via $\tilde{E}_j = \sum_{k=1}^m \frac{\partial \psi_k}{\partial \tilde{x}_j} \frac{\partial}{\partial x_k}$, we can rewrite it as

$$(\nabla_{\tilde{E}_j} V)(x) = \left( P \left( \sum_{i=1}^m \sum_{k=1}^m \frac{\partial \psi_k}{\partial \tilde{x}_j} \frac{\partial \bar{v}_i}{\partial x_k} \frac{\partial}{\partial x_i} \right) \right)(x) = \sum_{i=1}^m \left( P_x \frac{d\bar{v}}{dx} \frac{d\psi}{d\tilde{x}} \right)_{ij} \frac{\partial}{\partial x_i}. \tag{109}$$

On the other hand,

$$\tilde{E}^j = \sum_{k=1}^d g^{kj} \tilde{E}_k = \sum_{i=1}^m \sum_{k=1}^d \left( \frac{d\psi}{d\tilde{x}}^\top \frac{d\psi}{d\tilde{x}} \right)_{kj}^{-1} \frac{\partial \psi_i}{\partial \tilde{x}_k} \frac{\partial}{\partial x_i} = \sum_{i=1}^m \left( \frac{d\psi}{d\tilde{x}} \left( \frac{d\psi}{d\tilde{x}}^\top \frac{d\psi}{d\tilde{x}} \right)^{-1} \right)_{ij} \frac{\partial}{\partial x_i}. \tag{110}$$

Since $g$ is the induced metric, the summation over $j = 1, \cdots, d$ is equivalent to the Frobenius inner product $\langle \cdot, \cdot \rangle_F$ of the two $m \times d$ matrices

$$\sum_{j=1}^d \langle \nabla_{\tilde{E}_j} V, \tilde{E}^j \rangle_g = \left\langle P_x \frac{d\bar{v}}{dx} \frac{d\psi}{d\tilde{x}}, \frac{d\psi}{d\tilde{x}} \left( \frac{d\psi}{d\tilde{x}}^\top \frac{d\psi}{d\tilde{x}} \right)^{-1} \right\rangle_F \tag{111}$$

$$= \operatorname{tr} \left( P_x \frac{d\bar{v}}{dx} \frac{d\psi}{d\tilde{x}} \left( \frac{d\psi}{d\tilde{x}}^\top \frac{d\psi}{d\tilde{x}} \right)^{-1} \frac{d\psi}{d\tilde{x}}^\top \right) \tag{112}$$

$$= \operatorname{tr} \left( P_x \frac{d\bar{v}}{dx} P_x \right). \tag{113}$$

$\square$

**Proposition 2.** *If $V$ is the tangential projection matrix, then $(V \cdot \nabla_g) V = 0$.*

*Proof.* By definition,

$$(V \cdot \nabla_g) V = \sum_{j=1}^m V_j \nabla_g \cdot V_j. \tag{114}$$

Denote by the $j$'th column of $P_x$ by $(P_x)_{:j}$. Applying the resulting tangent vector to any smooth function $f$ (evaluated at $x$) and applying (12) gives

$$((V \cdot \nabla_g) V)(f)(x) = \sum_{j=1}^{m} (\nabla_g \cdot V_j)(x) V_j(f)(x) \tag{115}$$

$$= \sum_{j=1}^{m} \operatorname{tr}\left( P_x \frac{\mathrm{d}(P_x)_{:j}}{\mathrm{d}x} P_x \right) \sum_{i=1}^{m} (P_x)_{ij} \frac{\partial f}{\partial x_i} \tag{116}$$

$$= \sum_{i=1}^{m} \sum_{j=1}^{m} (P_x)_{ij} \operatorname{tr}\left( P_x \frac{\mathrm{d}(P_x)_{:j}}{\mathrm{d}x} P_x \right) \frac{\partial f}{\partial x_i}. \tag{117}$$

That is, the resulting tangent vector's coefficients correspond to the tangential projection of the vector

$$\begin{bmatrix} \operatorname{tr}\left( P_x \frac{\mathrm{d}(P_x)_{:1}}{\mathrm{d}x} P_x \right) \\ \vdots \\ \operatorname{tr}\left( P_x \frac{\mathrm{d}(P_x)_{:m}}{\mathrm{d}x} P_x \right) \end{bmatrix} \tag{118}$$

which we claim is orthogonal to the tangential linear subspace.

To prove the claim, we first note that we can rewrite $P_x$ as

$$P_x = I - n_x n_x^\top \tag{119}$$

where $n_x$ is of type $\mathbb{R}^{m \times (m-d)}$, and the column vectors of $n_x$ are orthonormal, and orthogonal to the tangential linear subspace; that is to say, $P_x n_x = 0$. Using this representation, we can write the Jacobian as

$$\left( \frac{\mathrm{d}(P_x)_{:j}}{\mathrm{d}x} \right)_{kl} = - \sum_{r=1}^{m-d} \frac{\partial}{\partial x_l} (n_x)_{kr} (n_x)_{jr} \tag{120}$$

$$= - \sum_{r=1}^{m-d} (n_x)_{jr} \frac{\partial}{\partial x_l} (n_x)_{kr} + (n_x)_{kr} \frac{\partial}{\partial x_l} (n_x)_{jr}. \tag{121}$$

Now multiplying by the projection matrix from both sides gives

$$P_x \frac{\mathrm{d}(P_x)_{:j}}{\mathrm{d}x} P_x = - \sum_{r=1}^{m-d} (n_x)_{jr} P_x \begin{bmatrix} \nabla_x (n_x)_{1r}^\top \\ \vdots \\ \nabla_x (n_x)_{mr}^\top \end{bmatrix} P_x + \underbrace{P_x (n_x)_{:r}}_{0} \nabla_x (n_x)_{jr}^\top P_x. \tag{122}$$

Lastly, let

$$\tau_r = \operatorname{tr}\left( P_x \begin{bmatrix} \nabla_x (n_x)_{1r}^\top \\ \vdots \\ \nabla_x (n_x)_{mr}^\top \end{bmatrix} P_x \right) \tag{123}$$

which means (118) is simply

$$- \sum_{r=1}^{m-d} (n_x)_{:r} \tau_r. \tag{124}$$

This implies the claim is true, since this is nothing more than a linear combination of the column vectors of $n_x$, which is orthogonal to the tangential linear subspace.

$$\square$$

**Theorem 3** (**Marginally equivalent SDEs**). *For $\lambda \leq 1$, the marginal distributions of $X_{T-s}$ and $Y_s$ of the processes defined as below*

$$\mathrm{d}Y = \left(U_0 - \frac{\lambda}{2}\nabla_g \log q\right)\mathrm{d}s + \sqrt{1-\lambda}V \circ \mathrm{d}\hat{B}_s \qquad\qquad Y_0 \sim q(\cdot, 0) \qquad (20)$$

$$\mathrm{d}X = \left(\left(1 - \frac{\lambda}{2}\right)\nabla_g \log q - U_0\right)\mathrm{d}t + \sqrt{1-\lambda} \circ V\mathrm{d}\hat{B}_t \qquad X_0 \sim q(\cdot, T) \qquad (21)$$

*both have the density $q(\cdot, s)$. In particular, $\lambda = 1$ gives rise to an equivalent ODE.*

*Proof.* We work with the derivation version of (14):

$$\mathrm{d}Y = U_0\,\mathrm{d}t + V \circ \mathrm{d}\hat{B}_s, \qquad\qquad (125)$$

That is, $U_0(f) = \sum_k (Pr)_k \frac{\partial}{\partial \tilde{x}_k} f \circ \psi$, and $V$ is the tangential projection. The marginal density $q$ follows the Fokker-Planck PDE

$$\partial_s q = -\nabla_g \cdot (qU_0) + \frac{1}{2}\sum_{k=1}^m \nabla_g \cdot \left((\nabla_g \cdot (qV_k))\, V_k\right) \qquad (126)$$

$$= -\nabla_g \cdot (qU_0) + \frac{1}{2}\sum_{k=1}^m \nabla_g \cdot \left((V_k(q) + q\nabla \cdot V_k)\, V_k\right) \qquad (127)$$

$$= -\nabla_g \cdot (qU_0) + \frac{1}{2}\sum_{k=1}^m \nabla_g \cdot (V_k(q)V_k) \qquad (128)$$

$$= -\nabla_g \cdot (qU_0) + \frac{1}{2}\sum_{k=1}^m \nabla_g \cdot (qV_k(\log q)V_k) \qquad (129)$$

$$= -\nabla_g \cdot (qU_0) + \frac{1}{2}\nabla_g \cdot (q\nabla_g \log q)\,, \qquad (130)$$

where we have used the product rule, and Proposition 2, the chain rule, and Proposition 4.

For $\lambda \leq 1$, we can rearrange the Fokker-Planck and get

$$\partial_s q = -\nabla_g \cdot \left(q\left(U_0 - \frac{\lambda}{2}\nabla_g \log q\right)\right) + \frac{1-\lambda}{2}\nabla_g \cdot (q\nabla_g \log q)\,, \qquad (131)$$

which is the Fokker-Planck equation of the process (20).

To construct a reverse process inducing the same family of marginal densities, we mirror the diffusion term around 0:

$$\partial_s q = -\nabla_g \cdot \left(q\left(U_0 - \left(1 - \frac{\lambda}{2}\right)\nabla_g \log q\right)\right) - \frac{1-\lambda}{2}\nabla_g \cdot (q\nabla_g \log q) \qquad (132)$$

Now we apply a change of variable of time via $p(x, t) = q(x, T - t)$, which means $\partial_t p = -\partial_s q|_{s=T-t}$ and thus

$$\partial_s p = -\nabla_g \cdot \left(q\left(\left(1 - \frac{\lambda}{2}\right)\nabla_g \log q - U_0\right)\right) + \frac{1-\lambda}{2}\nabla_g \cdot (q\nabla_g \log q)\,, \qquad (133)$$

which is the Fokker-Planck of (21).

$\square$

**Theorem 4** (**Score matching equivalency**). *For $\lambda < 1$, let $\mathcal{E}_\lambda^\infty$ denote the Riemannian CT-ELBO of the generative process (21), with $\nabla_g \log q$ replaced by an approximate score $S_\theta$, and*

*Proof.* Approximating $\nabla_g \log q$ in (21) using $S_\theta$ and plugging it in (5) and (20) in (9), we get

$$V_0 = \left(1 - \frac{\lambda}{2}\right) S_\theta - U_0 \tag{134}$$

$$\sqrt{1-\lambda} Va = (1-\lambda) S_\theta + \frac{\lambda}{2}\left(S_\theta - \nabla_g \log q\right). \tag{135}$$

Also, as we only need to focus on the tangential components of $a$, note that

$$\|Va\|_g^2 = \left\langle \sum_k V_k a_k, \sum_{k'} V_{k'} a_{k'} \right\rangle_g \tag{136}$$

$$= \sum_{kk'} a_k a_{k'} \langle V_k, V_{k'} \rangle_g \tag{137}$$

$$= \sum_{kk'} a_k a_{k'} \left\langle \sum_j P_{jk} E_j, \sum_{j'} P_{j'k'} E_{j'} \right\rangle_g \tag{138}$$

$$= \sum_{kk'jj'} a_k a_{k'} P_{jk} P_{j'k'} \langle E_j, E_{j'} \rangle_g \tag{139}$$

$$= \sum_{kk'j} a_k a_{k'} P_{jk} P_{jk'} = \|Pa\|_2^2, \tag{140}$$

where $E_j$ denote the ambient space Euclidean derivation $\frac{\partial}{\partial x_j}$.

Thus, we have

$$\frac{1}{2}\|Pa\|_2^2 = \frac{1}{2(1-\lambda)}\left[(1-\lambda)^2 \|S_\theta\|_g^2 + (1-\lambda)\lambda\langle S_\theta, S_\theta - \nabla_g \log q\rangle_g + \frac{\lambda^2}{4}\|S_\theta - \nabla_g \log q\|_g^2\right]$$

$$= \left(1 - \frac{\lambda}{2}\right)\frac{1}{2}\|S_\theta\|_g^2 + \frac{\lambda}{2}\left(\frac{1}{2}\|S_\theta\|_g^2 - \langle S_\theta, \nabla_g \log q\rangle_g\right) + \frac{\lambda^2}{4(1-\lambda)}\frac{1}{2}\|S_\theta - \nabla_g \log q\|_g^2$$

$$\nabla \cdot V_0 = \left(1 - \frac{\lambda}{2}\right)\nabla \cdot \left(S_\theta - \left(\frac{2}{2-\lambda}\right)U_0\right)$$

Summing up these two parts gives us $\mathcal{E}_\lambda^\infty$. Taking the expectation over $q(\cdot, 0)$ and applying the divergence theorem give us the desired identity. $\square$

# C  Manifolds

We provide some background on the manifolds used in this paper.

## C.1  Spheres and tori

Spheres are defined as submanifolds in an Euclidean space of points with unit Euclidean norm. Precisely, an $d$-sphere is $\mathbb{S}^d = \{x \in \mathbb{R}^{d+1} : \|x\|_2 = 1\}$. Therefore the ambient space dimensionality of a $d$ sphere is $m = d + 1$. Tori are products of 1-spheres (or circles); that is $\mathbb{T}^d = \Pi_{i=1}^d \mathbb{S}^1$. Naturally, we can embed a $d$-torus in a $m = 2d$-dimensional ambient space.

**Tangential projection**    Without loss of generality, we derive the orthogonal projection to the tangent space of spheres. The tangential projection of tori is just the same linear operator applied to $d$ $\mathbb{R}^2$ vectors independently.

To derive the tangential project, we note that any any incremental change in $x$, denoted by $\mathrm{d}x$, will need to leave the norm $\|x\|_2$ unchanged. That is,

$$\mathrm{d}\|x\|_2^2 = 2x\,\mathrm{d}x = 0. \tag{141}$$

This means $x$ is normal to the tangential linear subspace. We can find the orthogonal projection onto the tangent space by subtracting the normal component, via $P_x = I - \frac{xx^\top}{\|x\|_2^2}$.

**Closest-point projection**    The closest-point projection onto the sphere is $\pi(x) = \frac{x}{\|x\|_2}$. One can verify this is the point on $\mathbb{S}^d$ that minimizes the Euclidean distance from $x \in \mathbb{R}^{d+1} \setminus \{0\}$.

## C.2    Hyperbolic spaces

We work with the Lorentzian model of the hyperbolic manifold, which, like the $d$-spheres, is a $d$-manifold embedded in $\mathbb{R}^{d+1}$, defined as

$$\mathbb{H}_K^d := \{x = (x_0, \dots, x_d) \in \mathbb{R}^{d+1} : \langle x, x \rangle_\mathcal{L} = 1/K,\ x_0 > 0\}, \tag{142}$$

where $K < 0$ is the curvature of the manifold, and $\langle \cdot, \cdot \rangle_\mathcal{L}$ is the Lorentzian inner product

$$\langle x, y \rangle_\mathcal{L} = -x_0 y_0 + x_1 y_1 + \cdots + x_n y_n. \tag{143}$$

In our experiments, $K = -1$.

The $d+1$-dimensional Euclidean space endowned with the Lorentzian inner product $(\mathbb{R}^{d+1}, \langle \cdot, \cdot \rangle_\mathcal{L})$ is known as the Minkowski space. The Lorentz inner product is in general indefinite. Therefore, technically it is not an inner product. But it is positive definite when restricted to $\mathbb{H}_K^d$, and as a result induces a valid Riemannian metric $g_\mathcal{L}$. Equation (12), however, relies on the Euclidean geometry of the ambient space. Therefore, we model the density $p_\mathcal{E}$ associated with the metric tensor $g_\mathcal{E}$ induced by the regular Euclidean inner product. That is, $p_\mathcal{E}$ is a probability density of the manifold $(\mathbb{H}_K^d, g_\mathcal{E})$. Note that all the data points still lie on the same topological space $\mathbb{H}_K^d$, and the density can be translated via $p_\mathcal{E} = p_\mathcal{L} \sqrt{\frac{|G_\mathcal{L}|}{|G_\mathcal{E}|}}$, where $G_\mathcal{L}$ and $G_\mathcal{E}$ are the components of the matrix $g_\mathcal{L}$ and $g_\mathcal{E}$, and $p_\mathcal{L}$ is the actual density on the Hyperbolic manifold $(\mathbb{H}_K^d, g_\mathcal{L})$. This change-of-volume relation implies instead of maximizing the likelihood $\log p_\mathcal{L}$, we can simply maximize $\log p_\mathcal{E}$.

Alternatively, one can also compute the Riemannian divergence wrt the metric $g_\mathcal{L}$ using the internal coordinates, as is done in (Lou et al., 2020). In this case, the learned density will be the actual density $p_\mathcal{L}$ on the hyperbolic manifold.

**Tangential projection**    Similar to the spheres, we analyze the contribution of the differential $\mathrm{d}x$.

$$\mathrm{d}\langle x, x \rangle_\mathcal{L} = 2n_x\,\mathrm{d}x = 0, \tag{144}$$

where $n_x = (-x_0, x_1, \dots, x_d)$ is the normal vector. Subtracting the normal contribution gives rise to the tangential projection $P_x = I - \frac{n_x n_x^\top}{\|n_x\|_2^2}$.

Note that this is different from the usual "Lorentz" orthogonal projection $P_x^\mathcal{L}(u) = u - \frac{\langle x, u \rangle_\mathcal{L}}{\langle x, x \rangle_\mathcal{L}}x$ (Ratcliffe, 1994); the latter is not orthogonal in the Euclidean inner product.

**Closest-point projection**    We first derive the closest-point projection wrt the Lorentz inner product. For any $x \in \{x' : \langle x', x' \rangle_\mathcal{L} < 0\}$,

$$\pi(x) = \underset{y \in \mathbb{H}_K^d}{\arg\min} \|x - y\|_\mathcal{L}^2, \tag{145}$$

where $\|x\|_\mathcal{L} := \sqrt{\langle x, x \rangle_\mathcal{L}}$ is the Lorentz norm. To deal with the constraint $y \in \mathbb{H}_K^d$, we can introduce the Lagrange multiplier $\lambda$, and find the stationary point of the function

$$\|x - y\|_\mathcal{L}^2 + \lambda(\langle y, y \rangle_\mathcal{L} - 1/K). \tag{146}$$

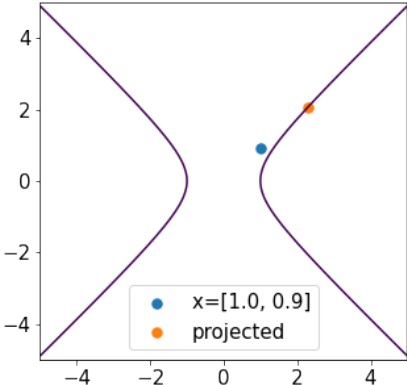

Figure 7: Closest-point projection of the point $(1.0, 0.9)$ onto the Hyperbolic manifold $\mathbb{H}^1_{-1}$ in the Lorentz norm. This projection is clearly not the closest one in Euclidean distance.

Taking the gradient wrt $y$ and setting it to be zero yield

$$-2n_{x-y} + 2\lambda n_y = 0 \iff y = \frac{1}{\lambda + 1}x. \tag{147}$$

On the other hand, $y \in \mathbb{H}^d_K$, which means $\lambda + 1 = \sqrt{K \|x\|_\mathcal{L}}$, and therefore

$$\pi(x) = \frac{x}{\sqrt{K \|x\|_\mathcal{L}}}. \tag{148}$$

This projection, however, is not the closest-point projection in Euclidean distance in general, as depicted in Figure 7. This is contrary to the claim made by Skopek et al. (2019). In fact, following the same derivation (using Euclidean distance in place of the Lorentz norm in (145)), we would end up with a Lagrange multiplier that cannot be analytically solved, as it involves solving a root finding problem.

This projection, albeit not the shortest one in Euclidean distance, is still a valid projection. We use it in numerical integration to simulate the dynamics.

### C.3 Orthogonal groups

The orthogonal groups are defined as $O(n) = \{X \in \mathbb{R}^{n \times n} : X^\top X = XX^\top = I\}$. The determinant of $X$ is either $1$ or $-1$. The subgroup with determinant $1$ is called the special orthogonal group, denoted by $SO(n)$. Naturally, $\mathbb{R}^{n \times n}$ is an ambient space of the orthogonal groups.

**Tangential projection** Following the differential analysis,

$$d(XX^\top) = X dX^\top + dX X^\top = 0. \tag{149}$$

That is, $dXX^\top$ is skew-symmetric. Denote the set of skew-symmetric matrices by $\text{Skew}_n = \{X \in \mathbb{R}^{n \times n} : X^\top = -X\}$.

Let $U$ be an arbitrary matrix in $\mathbb{R}^{n \times n}$. We want to project it orthogonally onto $\mathcal{T}_X O(n)$. The orthogonal projection needs to be the closest-point projection onto the subspace. We can use the Frobenius norm to induce the Euclidean distance metric over the entries of the matrix. Then finding the closest-point projection $V$ of $U$ amounts to finding the stationary point of

$$\|U - V\|_F^2 + \langle \Lambda, XV^\top + VX^\top \rangle_F, \tag{150}$$

where $\Lambda$ is the Lagrange multiplier. Taking the gradient wrt $V$ yields

$$\frac{d}{dV} \langle U - V, U - V \rangle_F + \langle \Lambda, XV^\top + VX^\top \rangle_F = \frac{d}{dV} \text{tr}((U-V)^\top(U-V) + \Lambda^\top(XV^\top + VX^\top))$$

$$= \frac{d}{dV} \text{tr}(-2U^\top V + V^\top V + VX^\top \Lambda + XV^\top \Lambda)$$

$$= -2U + 2V + \Lambda^\top X + \Lambda X.$$

Equating the last step with $0$ yields

$$V = U + \frac{\Lambda + \Lambda^\top}{2} X. \tag{151}$$

Since $V$ needs to satisfy $XV^\top + VX^\top = 0$, we have

$$XV^\top + VX^\top = XU^\top + \frac{\Lambda + \Lambda^\top}{2} + UX^\top + \frac{\Lambda + \Lambda^\top}{2} \tag{152}$$

$$= XU^\top + UX^\top + \Lambda + \Lambda^\top = 0, \tag{153}$$

which means

$$\Lambda + \Lambda^\top = -XU^\top - UX^\top. \tag{154}$$

Substituting this into (151) yields

$$V = \frac{U - XU^\top X}{2}. \tag{155}$$

That is, $P_X(U) = \frac{U - XU^\top X}{2}$ for orthogonal groups.

**Closest-point projection**  Again, using the Lagrange multiplier $\Lambda$ for the constraint that the projection $M$ of $X$ should satisfy $M^\top M = I$, we try to find the stationary point of the following quantity

$$\|M - X\|_F^2 + \langle \Lambda, M^\top M - I \rangle_F. \tag{156}$$

Equating the gradient wrt $M$ with $0$ gives

$$\frac{\mathrm{d}}{\mathrm{d}M} \langle M - X, M - X \rangle_F + \langle \Lambda, M^\top M - I \rangle_F = \frac{\mathrm{d}}{\mathrm{d}M} \mathrm{tr}((M - X)^\top (M - X) + (M^\top M - I)^\top \Lambda)$$

$$= \frac{\mathrm{d}}{\mathrm{d}M} \mathrm{tr}(M^\top M - 2X^\top M + M^\top M \Lambda)$$

$$= 2M - 2X + M\Lambda + M\Lambda^\top = 0,$$

which means

$$M = 2X(2I + \Lambda + \Lambda^\top)^{-1}. \tag{157}$$

Since $M$ is orthogonal, we have

$$M^\top M = 4(2I + \Lambda + \Lambda^\top)^{-T} X^\top X (2I + \Lambda + \Lambda^\top)^{-1} = I, \tag{158}$$

which means

$$4X^\top X = (2I + \Lambda + \Lambda^\top)^2. \tag{159}$$

Let $X = UDV^\top$ be the singular value decomposition of $X$. Then

$$2VDV^\top = 2I + \Lambda + \Lambda^\top. \tag{160}$$

Substituting this into (157), we get

$$M = XVD^{-1}V^\top = UDV^\top VD^{-1}V^\top = UV^\top. \tag{161}$$

That is, $\pi(X) = UV^\top$ for orthogonal groups, where $U, V$ are the left and right singular matrices of $X$.

| Manifold | Activation | Hidden layers | Embedding size | ActNorm first |
|---|---|---|---|---|
| Sphere | Sine | 5 | 512 | False |
| Tori | Swish | 4 | 256 | False |
| Hyperbolic | Swish | 2 | 512 | True |
| Orthogonal group | Swish | 256 | 256 | False |

Table 3: The variational function $a$ network architectures for different manifolds in our experiments.

| Manifold | Optimizer | Learning rate | $\beta_1$ | $\beta_2$ | Scheduler |
|---|---|---|---|---|---|
| Sphere | Adam | $2e-4$ | 0.9 | 0.999 | Cosine |
| Tori | Adam | $3e-4$ | 0.9 | 0.999 | None |
| Hyperbolic | Adam | $5e-4$ | 0.9 | 0.999 | None |
| Orthogonal group | Adam | $1e-3$ | 0.9 | 0.999 | None |

Table 4: Optimization hyperparameters for experiments on different manifolds

# D   Experimental details

## D.1   Architecture

In our experiments, we parameterize the $a$ network as a multi-layer perceptron (MLP) with either the sinusoidal or the swish activation function. For the hyperbolic experiments, the first layer of the MLP has an additional ActNorm layer (Kingma & Dhariwal, 2018) which we find adds extra numerical stability. The ActNorm layer is initialized before training with one batch such that its output has a mean of zero and a standard deviation of one. In an analogous manner to training the MLP the ActNorm parameters are updated via backpropagation. For the orthogonal group experiments we flatten the input matrix into a vector before passing it to the MLP. The details of our various model are given in Table 3. For our importance sampler which is used to represent a differentiable distribution over $[0, T]$, we use a deep sigmoidal flow (Huang et al., 2018) (without the final logit activation) followed by a fixed scaling flow, which represents the range $[0, T]$. We disconnect the gradient from the numerical solver to save compute; *i.e.* $Y_s$ is not differentiable. This would result in slightly biased gradient updates for minimizing the variance of the importance estimator, but we still observe substantial reduction in variance (see Figure 2). Finally, we use *PyTorch* (Paszke et al., 2019) as our deep learning framework.

**Computational Resources**. We run all of our experiments either on a single NVIDIA Tesla V100 or a single NVIDIA Quadro RTX 8000 GPU for a maximum of 30 hours.

## D.2   Optimization

We use the Adam (Kingma & Ba, 2015) optimizer to train the $a$ network. The learning rate and momentum parameters used for each manifold is mentioned in the Table 4. For the sphere experiments, we slowly decrease the learning rate during training using a cosine scheduler. For optimization of our importance sampler, we use Adam with a fixed learning rate of $0.01$. We update the importance sampler every $500$ steps of our training loop for the $a$ network. Lastly, to optimize our mixture of power spherical distributions for the tori experiments we use Adam with a learning rate of $0.03$ with $\beta_1 = 0.9$ and $\beta_2 = 0.999$.

## D.3   KELBO

The gap between the exact likelihood of the data given the model, *i.e.* $\log p(x)$, and the Riemannian CT-ELBO may be large. This evaluation gap makes empirical validation of the models using the Riemannian CT-ELBO imprecise. We acquire a tighter lower bound by using $K > 1$ samples and importance sampling similar to Burda et al. (2015). In details, we know from (90) that:

$$\log p(x, t) = \log \mathbb{E}_{\mathbb{Q}} \left[ \frac{d\mathbb{P}}{d\mathbb{Q}} \cdot p_0\left(Y_t\right) \exp\left( -\int_0^T \nabla_g \cdot \left( V_0 - \frac{1}{2}(V \cdot \nabla_g)V \right) ds \right) \middle| Y_0 = x \right].$$

| Manifold | Integration steps during training |
|---|---|
| Sphere | 100 |
| Tori | 1000 |
| Hyperbolic | 100 |
| Orthogonal group | 100 |

Table 5: Details of training integration.

| rtol | atol | minimum step size |
|---|---|---|
| $1e-3$ | $1e-3$ | $1e-5$ |

Table 6: Configuration of the adaptive step size integration used during evaluation.

We rewrite this as:

$$\log p(x,t) = \log \mathbb{E}_{\mathbb{Q}} L(Y),$$

where $L(Y)$ is defined to be:

$$\frac{\mathrm{d}\mathbb{P}}{\mathrm{d}\mathbb{Q}} \cdot p_0(Y_t) \exp\left(-\int_0^T \nabla_g \cdot \left(V_0 - \frac{1}{2}(V \cdot \nabla_g)V\right) \mathrm{d}s\right).$$

Then by Jensen's inequality, we have that:

$$\log p(x,t) = \log \mathbb{E}_{\mathbb{Q}} L(Y) = \log \mathbb{E}_{\mathbb{Q}} \sum_{i=1}^{K} \frac{1}{K} L(Y^i) \geq \mathbb{E}_{\mathbb{Q}} \log \sum_{i=1}^{K} \frac{1}{K} L(Y^i),$$

where $Y^i$s are *i.i.d.* trajectories sampled from $\mathbb{Q}$. We call this new lower bound KELBO. Note that this is a tighter lower bound because we can write:

$$\text{KELBO} = \mathbb{E}_{\mathbb{Q}} \log \sum_{i=1}^{K} \frac{1}{K} L(Y^i) \geq \mathbb{E}_{\mathbb{Q}} \sum_{i=1}^{K} \frac{1}{K} \log L(Y^i) = \mathbb{E}_{\mathbb{Q}} \log L(Y) = \text{Riemannian CT-ELBO}.$$

In fact, this lower bound increases monotonically to the true likelihood as $K \to \infty$. We use KELBO with $K = 100$ for evaluating all of our models. We have experimented with the $K$ to be up to 1000 and found out the results stop changing much for $K > 100$.

### D.4 Numerical integration of the SDEs

During training and evaluation, we numerically integrate the SDE on each respective manifold using the Stratonovich-Heun method as described in Burrage et al. (2004). Each iteration is followed by the closest-point projection (in the case of $\mathbb{H}_K^d$, we use the closest-point project wrt the Lorentz inner product). The number of integration steps for each manifold during training is reported in Table 5.

During evaluation, as described in D.3, we numerically integrate the data from $s = 0$ to $s = T$, and the Itô integral involved in the KELBO is approximated using the Euler-Maruyama scheme (note that the dynamics is still generated using Stratonovich-Heun). As computing KELBO requires forward passes through the $a$ network, it may not be as smooth as just integrating the inference SDE. Therefore, we use an adaptive step size for integration. We adapted the *torchsde* library (Kidger et al., 2021; Li et al., 2020) to calculate errors and adapt the step size accordingly. The error tolerance and minimum step size used in integration for all the experiments are reported in Table 6. Also, for plotting densities we use the exact log likelihood of the equivalent ODE. To numerically integrate the ODE for computing the exact likelihood, we use the default dopri5 solver from the *torchdiffeq* library (Chen et al., 2018b, 2021). Finally, we use *cartopy* (Met Office, 2010 - 2015), *matplotlib* (Hunter, 2007), and *plotly* (Inc., 2015) for visualization.