# OpenReview forum: "Riemannian Diffusion Models"
_NeurIPS.cc/2022/Conference — NeurIPS 2022 Accept_

### Official Review · Reviewer_KTty · 2022-06-28

**Rating:** 7
**Confidence:** 4
**Soundness:** 4 excellent
**Presentation:** 4 excellent
**Contribution:** 3 good

**Summary:**

The paper derives a variational framework for likelihood estimation of diffusion models on Riemannian manifolds. This is based on prior work on continuous time ELBOs and score matching. The authors generalize this to Riemannian manifolds and address computational issues including fast computation of the Riemannian divergence. The methods are experimentally demonstrated on several manifolds.

**Questions:**

* line 172-173: I didn't get what is meant by "is evaluated only once as opposed to estimating the ELBO by integrating over the entire path of Y_s". What is evaluated only once? The entire sample path is still used?

**Limitations:**

yes

**Strengths And Weaknesses:**

Strengths:
* well-written and technically sound paper
* addresses an important class of problems
* the presented techniques are mathematically interesting and non-trivial
* the paper couples stochastic process theory, variational inference and geometry

Weaknesses:
* the CT-ELBO was derived in earlier paper in the Euclidean setting. To some extend, the extension to the geometric context is not groundbreaking. However, it is in no way trivial either

In general, I found the paper a very interesting read.

---

> ### Author Response · Authors · 2022-07-30
> **Response to Reviewer KTty**
>
> Thank you, we are thrilled that you like our work!
>
> We appreciate your comment regarding the CT-ELBO being derived in an earlier work. We wish to highlight that the generalization to the Riemannian setting is a much more involved and difficult endeavor and one easily recovers the original CT-ELBO if the manifold is flat (e.g. Euclidean space). With this in mind our proposed model is as a result more encompassing—i.e. we can cover many more manifolds of interest while providing a numerically stable and efficient way to do generative modelling by optimizing the marginal likelihood.
>
> Regarding line 172-173, this refers to evaluating the “$a$” function on one single $Y_s$, as opposed to numerically carrying out the integration in (15). Obtaining $Y_s$ is relatively cheap since its dynamics is smoother, following (14). We will make this point clearer.

---

### Official Review · Reviewer_JCSo · 2022-07-11

**Rating:** 7
**Confidence:** 3
**Soundness:** 4 excellent
**Presentation:** 3 good
**Contribution:** 3 good

**Summary:**

This paper extends the application of continuous-time diffusion models into Riemannian manifold. The paper proposes a new method of calculating the Riemannian divergence for computing the Riemannian lower variational bound objective function.  In particular, the authors also makes an insightful connection with Riemannian score matching.

**Questions:**

- Why is there only one baseline model comparison in the tori results specifically not having the comparison with the Riemannian Score-Based Model?
- In some tasks, SGM have performed better in certain image generation than diffusion models (e.g. CIFAR-10, CelebA 256x256). In this task, it seems that this model performs better than its score-based counter part. Is there any reason to believe that diffusion models should theoretically perform better in the Riemann space?
- Projection can be a difficult task, especially projecting onto a high dimensional complex plane. Does this have any detrimental effects on the training/sampling time?

**Limitations:**

The authors address their limitations (i.e. algorithmic decisions).

**Strengths And Weaknesses:**

Strengths:
- Overall, the paper was well organized and clearly written.
- The shadowed boxes on the salient equations help follow the derivations.
- The idea is a novel application/expansion of diffusion models as they have (to the extent of my knowledge) not have been explored on non-Euclidean spaces.
- The connection between score-based models (in the Riemannian case) is a welcoming and insightful addition.

Weaknesses:
- The experiments show SOTA results and seem very promising, however, there seems to be a lack of analysis on the results.
- Limited information on training/sampling complexity.

Additional Remarks:
- Pg. 4 Line 136 "showed that that..." typo

---

> ### Author Response · Authors · 2022-07-30
> **Response to Reviewer JCSo**
>
> We would like to thank you for your positive feedback! We now address the key questions in detail below:
>
> > SOTA analysis and comparison to Riemannian score based model
>
> The reviewer makes an astute observation regarding comparison to Riemannian score based models. We would like to highlight the principal difference from the previous SOTA (which is the Riemannian score based generative model, RSM) and our proposed Riemannian Diffusion Model (RDM) is the fact that we directly maximize a lower bound on the marginal likelihood. This allows us to train an importance sampler to reduce variance of the time integral estimation in the ELBO. RSM’s, on the other hand, have a fixed inference noisy schedule which does not necessarily maximize the likelihood. The effect of variance reduction is demonstrated by Figure 2. We observed the gain in performance from using trainable importance sampling to reduce estimation variance is consistent across different hyperparameter settings and random seeds.
>
> > Training/sampling complexity
>
> We appreciate the reviewer’s comment but we would politely remind the Reviewer that the full description of training and sampling is provided in appendix D. In terms of training time complexity, we analyze it as part of section 5.2, where we noticed that for training approximately ~10^2 integration steps are sufficient for accurate ELBO estimation. At inference time, as the likelihood computation involves a more non-linear drift (the variational degree of freedom “a”), we adopted adaptive step size methods from torchsde (https://github.com/google-research/torchsde). We mention these details in Appendix D.4. For sampling we adopted fixed step-size integration, although in practice we can also employ adaptive step size.
>
> > Tori baseline
>
> We acknowledge the reviewer’s concern but we believe RSM is concurrent work. Furthermore, the code for RSM was not publicly available and upon contacting the authors we were informed the code was not yet ready for release and consequently we did not have access to their code at the time of submission. We also emphasize that RDM complements the score-based framework, as RDM is purely likelihood-based and so can leverage variance reduction techniques to avoid manually designing the inference time scheduling (the noise amount injected at different steps). We only report the results with variance reduction. It would be equivalent to RSM with likelihood-weighting if we simply remove importance sampling and sample time uniformly.
>
>
> > SGM versus diffusion model
>
> This depends on what metric we use to evaluate the model. For (likelihood-based) diffusion models, as we directly optimize a lower bound on the marginal likelihood of the model, it is natural and expected to have better likelihood. This is also empirically verified in Song et al 2021 [1] and Huang et al 2021 [2]. However, different weighting of the score matching loss could lead to improved performance in terms of perceptual quality (such as FID or visual quality), but such metrics have not been developed for the types of non-Euclidean datasets we choose to benchmark on (i.e. manifold data).
>
>
> > Projection
>
> The projection involved in our divergence computation only requires tangential projection (which is also required for the closest-point projection method proposed by Rozen et al 2021 [3]). This is a linear projection from the ambient space to the linear subspace of the tangent plane and can be represented by an $m$-by-$m$ matrix, where m is the dimensionality of the ambient space. This matrix-vector multiplication in some cases can also be simplified -- for instance, we can carry out the tangential projection of $O(n)$ (the orthogonal group of $n$-by-$n$ matrices embedded in the $m=n^2$ dimensional space; see Appendix C.3) using n-by-n matrix-matrix multiplications (as opposed to flattening out the matrix and applying an $n^2$-by-$n^2$ matrix-vector multiplication). This type of simplification is often dependent on the specific structure of the manifold.
>
> [1] Song, Y., Durkan, C., Murray, I., & Ermon, S. (2021). Maximum likelihood training of score-based diffusion models. Advances in Neural Information Processing Systems, 34, 1415-1428.
>
> [2] Huang, C. W., Lim, J. H., & Courville, A. C. (2021). A variational perspective on diffusion-based generative models and score matching. Advances in Neural Information Processing Systems, 34, 22863-22876.
>
> [3] Rozen, Noam, et al. "Moser flow: Divergence-based generative modeling on manifolds." Advances in Neural Information Processing Systems 34 (2021): 17669-17680.

---

> > ### Comment · Reviewer_JCSo · 2022-08-09
> > **Thank you for your reply**
> >
> > Dear author, thank you for the clarification. I believe you have answered all of my concerns. I think this paper is very novel and important for the diffusion model community. I will raise my score.

---

### Official Review · Reviewer_6m1R · 2022-07-11

**Rating:** 5
**Confidence:** 2
**Soundness:** 2 fair
**Presentation:** 2 fair
**Contribution:** 2 fair

**Summary:**

The paper proposes a general diffusion model for Riemannian manifolds. It introduces a variational framework based on Riemannian Feynman-Kac representation and Giransov’s theorem, with which a Rimannian continuous-time ELBO is derived. To compute the resulting ELBO, this paper further suggests a QR decomposition-based method for scalable unbiased estimation. Besides, it also provides a variance reduction technique to approximate the Remannian ELBO objective.


**Questions:**

In Eq. (27) of Appendix, what’s the meaning of  \frac{\partial \varphi_{j}^{-1}}{\partial \tilde{x}_{i}}? If my understanding is right, \varphi_{j}^{-1} should be understood as (\iota \circ \varphi)_j^{-1}

Regarding the spherical dataset, it is not clear to me why the used data lie in a sphere? Maybe some explanation or references will help interested readers.

Small typos should be corrected. For example, Eq. (24) in Appendix, the v in LHS should be capitalized.


**Ethics Review Area:**

["I don’t know"]

**Limitations:**

It seems this paper does not discuss its limitations. It will be highly appreciated if the authors can address my concerns/questions mentioned above.

**Strengths And Weaknesses:**

Strength:

Based on SDE, this paper generalized the Euclidean diffusion model into a Riemannian one. The generalization seems to enjoy a solid mathematical background.

The experiments demonstrate the effectiveness of the proposed method on the data that are on spherical manifolds, product of spherical manifolds, hyperbolic spaces and orthogonal groups.

Weakness:

This paper requests a lot of effort to understand its background and the proposed idea.

The presentation of background (Sec.2) is also very challenging to understand. For example, regarding Euclidean diffusion models, it might be better to further clarify how the diffusion model can be formulated as a fixed Markov chain that is presented in Line 24-26. Besides, it should be necessary to indicate the meanings of \mu and \sigma after Eq. (1). The same suggestion could be applied to other equations like Eq.(2).

As for the computation of the derived CT-ELBO, the suggested QR decomposition based method is an efficient approach. I cannot really understand why the paper further introduces a variance reduction technique to approximate the CT-ELBO. What is the major benefit of the used variance reduction over the QR decomposition based method? It seems that this paper does not study this benefit empirically.

---

> ### Author Response · Authors · 2022-07-30
> **Response to Reviewer 6m1R**
>
> We thank the reviewer for their time in reviewing our draft and providing useful feedback and questions which we now address.
>
>
> We understand that our proposed Riemannian Diffusion Model (RDM) may be a challenge to read, but we argue that this is a natural outcome given that our approach is built on distinct mathematical subfields such as stochastic processes, differential geometry, and approximate inference—all of which play important roles in our developed theory. We acknowledge the reviewer’s sentiment that it requires a lot of background knowledge to understand these topics but we provide a comprehensive and relatively self-contained treatment of all tools we borrow from differential geometry in Appendix A. Moreover, we would like to stress that both stochastic processes and differential geometry have been staple of the NeurIPS and broader machine learning community with the latter enjoying workshops in NeurIPS 2020 (DiffGeo4DL [1]) as well as the current NeurIPS 2022 conference (NeuReps [2]).
>
> We believe the tight and non-trivial interplay of our varied mathematical tools is a strength of the paper, as recognized by reviewer JCSo and reviewer KTty, who deemed the paper well written and organized despite its technical breadth. Having said that, we will incorporate your feedback to improve the manuscript. Specifically, we will try to point to references of prior work and book chapters whenever possible to make it more accessible to the more general machine learning community.
>
> We now address your specific questions below:
>
> > how the diffusion model can be formulated as a fixed Markov chain
>
> This refers to the fixed inference parameterization, eq (14), where the inference is a simple diffusion process. A diffusion process, by definition, is a continuous-time Markovian process with continuous sample paths. See Definition 2.2 of Pavliotis (2014) [3] for example.
>
>
> > $\mu$ and $\sigma$
>
> These are the drift and the diffusion coefficients of the diffusion process, which control the deterministic forces driving the evolution of the stochastic process and the amount of noise injected at each time step, respectively. We have updated the paper to include this.
>
>
> > Variance reduction
>
> We thank the reviewer for their question. Importance sampling is applied to the time integral “$\int \cdots ds$”, which is crucial to getting better performance as it reduces the variance of stochastic gradient by randomly choosing a time slice “s” to estimate the ELBO (as opposed to uniformly sampling between [0, T]). The benefit of importance sampling is empirically demonstrated by Figure 2.
>
> The chief goal of importance sampling (which is for estimating the “time integral”) is different from the computation (i.e. QR decomposition) and estimation (projected Hutchinson) of the “Riemannian divergence”. We emphasize that the QR decomposition method is exact, so there is no estimation involved. The only differences from other methods such as closest-point projection are with respect to “computation” (ours is $O(d)$ whereas closest-point is $O(m)$; recall m is the dimensionality of the ambient space and d is the dimensionality of the manifold) and “generality” (as we do not require an exact expression of the closest-point projection). The closest-point projection and the QR method we propose return the same value, up to a numerical rounding error which is negligible.
>
>
> > Eq (27)
>
> Yes, $\varphi^{-1}$ is indeed composed with the inclusion map. This is explained in line 533-535 (in current version, line 536-538) in the appendix.
>
> > Spherical dataset
>
> The reviewer raises an interesting question regarding our spherical dataset. We would like to politely remind the reviewer that the dataset consists of geospatial data such as floods and earthquakes which naturally occur on Earth which we model as a perfect sphere. Hence the data (the natural event) is a point on the spherical manifold which is then used towards our density estimation task. This is explained in Mathieu and Nickel 2020 [4] who introduced this dataset for generative modelling and we will add a pointer to this reference in the experimental section for further clarity.
>
> > Typo
>
> Thank you for spotting it! We will correct it.
>
> We thank the reviewer again for their time and feedback and we hope that our clarification points have thoroughly addressed any remaining questions and we hope the reviewer would kindly consider a fresh evaluation of our work given the main clarifying points outlined above.
>
>
> [1] https://sites.google.com/view/diffgeo4dl/
>
> [2] https://www.neurreps.org/
>
> [3] Pavliotis, G. A. (2014). Stochastic processes and applications: diffusion processes, the Fokker-Planck and Langevin equations (Vol. 60). Springer.
>
> [4] Mathieu, Emile, and Maximilian Nickel. "Riemannian continuous normalizing flows." Advances in Neural Information Processing Systems 33 (2020): 2503-2515.

---

> > ### Comment · Reviewer_6m1R · 2022-08-10
> > **The paper sounds good but should be further improved**
> >
> > Dear authors, thank you for the clarification. I would like to tune up my score. I believe it is very necessary to make the paper more accessible to the general machine learning community. In addition, it sounds like the main paper does not discuss its limitations as mentioned in my previous reviews. It would be appreciated if the authors can further discuss this regard.

---

> ### Author Response · Authors · 2022-08-09
> **Re: Rebuttal Acknowledgement**
>
> Dear Reviewer,
>
> Thank you for taking the time to review our paper and providing constructive feedback. We noticed that the reviewer acknowledged our rebuttal but we are wondering if our responses fully addressed the stated concerns. We would also like to note that Reviewer JCSo has increased their score from 6->7 after the rebuttal as well. As the end of the discussion phase is fast approaching we would love to answer any remaining doubts or questions the reviewer still has and incorporate any further feedback or suggestions for us to obtain a better evaluation. We thank you again for your time and commitment to the review process.

---

### Author Response · Authors · 2022-07-30
**Summary response**

We thank the reviewers for their feedback and helpful comments. We are heartened to hear that Reviewers JCSo and KTty found our paper “well organized and clearly written” despite the technical breadth (as recognized by all reviewers). We are further pleased that Reviewer JCSo found our method to be a novel extension of diffusion models to non-Euclidean spaces leveraging tight interplay between stochastic process theory, variational inference, and Riemannian geometry which—and as highlighted by Reviewer KTty, are both mathematically interesting and non-trivial. We also appreciate that Reviewer 6m1R found our proposed Riemannian Diffusion Models to enjoy a “solid mathematical background” while Reviewers JCSo and KTty evaluated our paper as being “technically sound”.  Finally, we thank Reviewer 6m1R for underlining that our experiments demonstrate the effectiveness of our proposed model. We now clarify the main shared concern regarding computing the CT-ELBO between the reviewers below and address reviewer specific questions in the individual responses.


**Computing the CT-ELBO**

We would like to clarify that the CT-ELBO (Eqn. 17) involves a time integral “$\int\dots ds$” and a Riemannian divergence “$\nabla_g \cdot$”. Section 3.3 (Variance reduction) addresses the time integral, while Section 3.1 (Computing Riemannian divergence) addresses the actual divergence computation. Applying the variance reduction technique is an advantage over Riemannian score based generative models (De Bortoli et. al) if our goal is to obtain better likelihood estimates, since we can view this as an estimation problem of the lower bound on the marginal likelihood. We understand that these details may not have been sufficiently clear in the original draft but we hope these facts fully address the questions by Reviewers 6m1R and JCSo.

*De Bortoli, Valentin, et al. "Riemannian score-based generative modeling." arXiv preprint arXiv:2202.02763 (2022)*

---

### Author Response · Authors · 2022-08-07
**Paper updated**

Dear reviewers,

We would like to thank you again for your endeavors in this reviewing process and your positive and constructive appraisal. We have updated the manuscript to incorporate your feedback and we hope we have fully addressed your questions. We would like to know if further clarification is needed, and are happy to engage in additional discussion if anything remains unclear.

---

### Meta-Review · Area_Chair_xkJC · 2022-08-23

**Recommendation:** Accept
**Confidence:** Certain

**Metareview:**

This paper presents a generalization of continuous-time diffusion models to Riemannian manifolds and derives a variational framework for likelihood estimation. The theoretical analysis is accompanied by numerical experiments.

Reviewers generally agree that the paper is novel, technically sound,  well-written, and would make a solid contribution to research on diffusion models.

Presentation: I would suggest that the presentation be improved to make it more accessible to the general ML community. Specific references should be given to concepts such as Riemannian divergence,  Divergence Theorem, etc , since these are not new.



**Award:**

No

---

### Decision · Program_Chairs · 2022-09-14

Accept